# *CDH2* mutation affecting N-cadherin function causes attention-deficit hyperactivity disorder in humans and mice

D. Halperin [1,10], A. Stavsky [2,10], R. Kadir[1], M. Drabkin [1], O. Wormser [1], Y. Yogev [1], V. Dolgin[1], R. Proskorovski-Ohayon [1], Y. Perez [1,3,4], H. Nudelman[5], O. Stoler[2], B. Rotblat [5], T. Lifschytz[6], A. Lotan [6], G. Meiri[7,8], D. Gitler [2] & O. S. Birk [1,9✉]

Attention-deficit hyperactivity disorder (ADHD) is a common childhood-onset psychiatric disorder characterized by inattention, impulsivity and hyperactivity. ADHD exhibits substantial heritability, with rare monogenic variants contributing to its pathogenesis. Here we demonstrate familial ADHD caused by a missense mutation in *CDH2*, which encodes the adhesion protein N-cadherin, known to play a significant role in synaptogenesis; the mutation affects maturation of the protein. In line with the human phenotype, CRISPR/Cas9-mutated knock-in mice harboring the human mutation in the mouse ortholog recapitulated core behavioral features of hyperactivity. Symptoms were modified by methylphenidate, the most commonly prescribed therapeutic for ADHD. The mutated mice exhibited impaired presynaptic vesicle clustering, attenuated evoked transmitter release and decreased spontaneous release. Specific downstream molecular pathways were affected in both the ventral midbrain and prefrontal cortex, with reduced tyrosine hydroxylase expression and dopamine levels. We thus delineate roles for *CDH2*-related pathways in the pathophysiology of ADHD.

[1] The Morris Kahn Laboratory of Human Genetics, National Institute for Biotechnology in the Negev and Faculty of Health Sciences, Ben-Gurion University of the Negev, Beer-Sheva, Israel. [2] Department of Physiology and Cell Biology, Faculty of Health Sciences and Zlotowski Center for Neuroscience, Ben-Gurion University of the Negev, Beer-Sheva, Israel. [3] Eli and Edythe Broad Center of Regeneration Medicine and Stem Cell Research, University of California, San Francisco, CA 94143, USA. [4] Department of Neurology, University of California, San Francisco, CA 94158, USA. [5] Department of Life Sciences and National Institute for Biotechnology in the Negev, Ben-Gurion University of the Negev, Beer-Sheva, Israel. [6] Biological Psychiatry Laboratory, Hadassah-Hebrew University Medical Center, Jerusalem, Israel. [7] Pre-School Psychiatry Unit, Soroka University Medical Center, Beer-Sheva, Israel. [8] Faculty of Health Sciences, Ben-Gurion University of the Negev, Beer-Sheva, Israel. [9] Genetics Institute, Soroka University Medical Center, Beer-Sheva, Israel. [10]These authors contributed equally: D. Halperin, A. Stavsky. ✉email: obirk@bgu.ac.il

Attention-deficit hyperactivity disorder (ADHD) is one of the most common childhood-onset neuropsychiatric conditions, characterized by a persistent pattern of inattention, impulsivity, and hyperactivity, with complications often continuing into adulthood[1]. Affected individuals have difficulties in higher-level executive functions, which are mediated by late-developing frontal-striatal-parietal and frontal-cerebellar neuronal networks. These mainly include motor and interference inhibition, working memory, sustained attention, and temporal information processing[2]. Although its etiology is not well defined, ADHD appears to have substantial heritability, and as such, it has been the focus of considerable genetic research, with growing evidence that rare monogenic variants may possess an essential role in its pathogenesis[3].

Here we describe three siblings of a consanguineous kindred presenting with severe ADHD, apparent as of early childhood. Through linkage analysis, whole-exome sequencing (WES), and biochemical studies, we identified a disease-associated homozygous missense mutation in *CDH2*, affecting proteolysis and maturation of the encoded N-cadherin adhesion protein, which is known to play a significant role in synaptogenesis, plasticity-induced long-term spine stabilization, and neurite outgrowth[4,5]. Notably, *CDH2* has an essential role in regulating the proliferation of dopaminergic progenitors within the limbic system, primarily the ventral midbrain and prefrontal cortex[6]. Through generation and analysis of mice homozygous for the human mutation in the mouse *CDH2* ortholog, we demonstrated hyperactivity and deficient sensorimotor integration in the mutant mice and delineated downstream physiological and molecular pathways mediating the phenotype, mainly alterations in synaptic properties and defects in dopamine neurotransmission. Thus, we identify the role of *CDH2* and its downstream pathways in the pathophysiology of ADHD.

## Results

**Clinical characterization.** Three siblings of consanguineous Bedouin pedigree (Fig. 1a) presented with severe ADHD, diagnosed as of early childhood. Patient II:6 was born at term following an uneventful pregnancy, whereas the non-identical twin patients II:3 and II:4 were born prematurely at 32 weeks (weight appropriate for gestational age; 1900 and 2200, respectively). By the age of three, all siblings presented with a similar manifestation of severe hyperactivity behavior, predominantly hyperactive/impulsive. By the age of four, all patients met the criteria for ADHD, as outlined in the DSM-5. Concisely, information about ADHD manifestations was collected from semi-structured interviews conducted with both parents. Additional information was obtained from observations, questionnaires, and supplementary assessments: Clinical Evaluation of Language Fundamentals 5th Edition[7] and Conner's Parent Rating Scales-Revised[8]. The following were excluded in all three patients: scoring below 80 on both the performance and the verbal scales of the WISC-III[9], psychosis, bipolar affective disorder, Tourette syndrome, multiple chronic tics, and a first-degree relative diagnosed with bipolar affective disorder and schizophrenia. All affected siblings were medication-free for 24 h prior to assessment and cognitive testing. Patients II:3 and II:6 (ages 11 and 7, respectively) reached normal developmental milestones, had no other comorbidities, and are completely normal in terms of intellect. Patient II:4 demonstrated mild developmental delay with autism spectrum disorder manifestations. Brain MRI (patient II:6, at 21 months) was normal. Blood pH, lactate, pyruvate, creatine, phosphokinase, and amino acids, as well as urinary organic acids, were within normal limits. Screening for congenital glycosylation defects, karyotype, and chromosomal microarrays were normal. All three patients were treated with stimulants, neuroleptics, and 3-Quinuclidinyl Benzilate. Notably, both parents and other siblings (including II:1 and

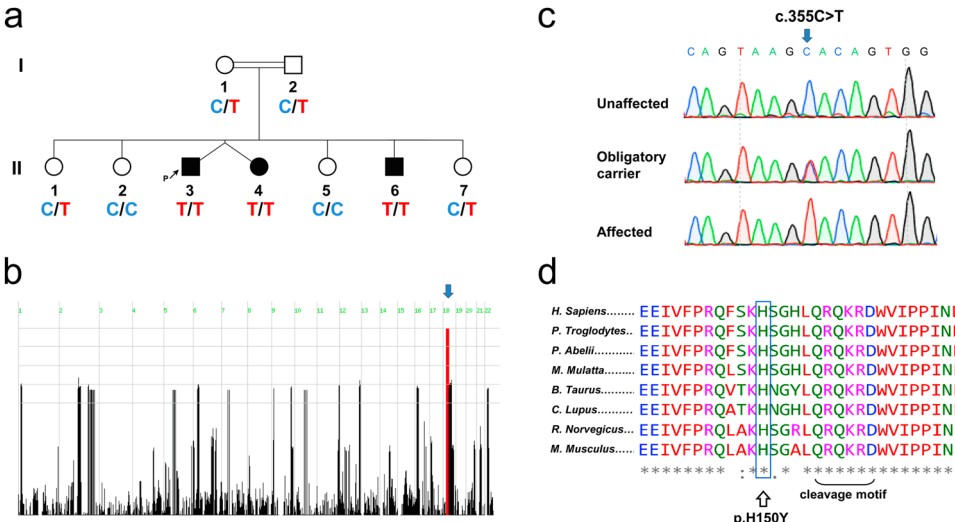

**Fig. 1 Pedigree and CDH2 mutation. a** Pedigree of the consanguineous kindred studied. Beneath each individual is the allele corresponding to the *CDH2* mutation. C and T denote the WT or mutant nucleotide, respectively. **b** Homozygosity scores. Genome-wide single nucleotide polymorphism (SNP) distributions were collected for all nine family members by bead-chip (>750 k/sample). The distribution of homozygous regions in the genome was determined using HomozygosityMapper (http://www.homozygositymapper.org/). Genomic regions are ordered by chromosome (green numbers). The blue arrow indicates the single homozygous locus on chromosome 18 shared by affected individuals. **c** Sanger sequencing. Through whole-exome sequencing, a single homozygous variant was found within the segregating locus: c.355 C > T in *CDH2*. *CDH2* sequencing results of an unaffected individual (II:2), an obligatory carrier (I:1), and an affected individual (II:3) are shown. **d** Protein MSA. To demonstrate evolutionary conservation within the vicinity of mutated p.H150Y residue, eight representative *CDH2* orthologs were selected for MSA. The recognition motif RXK/R-R is located approximately five residues downstream of the identified site of mutation. MSA, multiple sequence alignment.

II:7) were normal in terms of hyperactivity, intellect, and general health.

**Genetic analysis**. Linkage analysis, testing all nine family members, identified only one locus shared by the affected siblings: a ~11 Mb homozygous segment on chromosome 18 between SNPs rs11082423 and rs1480438 (Fig. 1b). Homozygosity mapping delineated this segment as the only homozygous disease-associated locus segregating as expected for autosomal recessive heredity within the studied kindred. WES data of individual II:3 were filtered for normal variants as described in Methods. A single homozygous variant was found within the locus: c.355 C > T; p.H150Y in *CDH2* (transcript variant 1; NM_001792.4). This variant, validated by Sanger sequencing (Fig. 1c), was found to segregate within the family as expected for autosomal recessive heredity; neither compound heterozygous nor other homozygous mutations were found to co-segregate within this locus. Screening of the variant in 400 ethnically matched controls identified a single carrier and no homozygous mutants. This mutation has not been reported in the Genome Aggregation Database (gnomAD), with only fifteen *CDH2* loss-of-function (LoF) variants (stop gain, frameshift, or essential splice site mutations) reported to date, none in a homozygous state. Multiple sequence alignment demonstrated the p.H150 residue to be highly conserved (Fig. 1d).

**CDH2 protein structural analysis**. *CDH2* encodes a 906 amino acid protein, neuronal cadherin (N-cadherin), that is broadly expressed in the brain. It is known to play an essential role in synaptogenesis, synapse function, plasticity-induced long-term spine stabilization, and cortical organization. Classical cadherins are initially synthesized bearing a prodomain, thought to limit adhesion during the early stages of biosynthesis, which is then endogenously cleaved within the Golgi apparatus[10]. N-cadherin protein modeling (Fig. 2a) revealed that the sequence linking the prodomain to the outermost extracellular cadherin domain is unstructured and can be found in variable conformational loops, enabling the anchoring of proteolytic enzymes, mainly furin protease[11]. Computational studies (Fig. 2b) demonstrated that the furin protease consensus cleavage site contains approximately 20 residues, harboring the recognition motif RXK/R-R[12], located five residues downstream to the p.H150Y mutation site. Using prediction tools for protein-peptide interactions, we demonstrated that the wildtype (WT) p.H150 residue is anchored within the catalytic pocket of the furin protease active-site, putatively contributing to its stable docking. In contrast, the mutant tyrosine sidechain, in a right rotamer conformation, is predicted to project perpendicularly, away from the catalytic pocket due to its non-polar, uncharged sidechain. Therefore, the mutation is predicted to interfere with the proteolysis and maturation of the protein.

**Decreased cleavage efficacy of the mutated p.H150Y peptide**. The proprotein convertase family of enzymes plays an important role in activating other proteins[13]. To test whether the p.H150Y mutation interferes with protein processing, we performed a biochemical peptide cleavage assay using furin protease, the prototypical proprotein convertase; WT and mutant 22 amino-acid peptides were synthesized (GL Biochem, Shanghai), harboring the aforementioned RXK/R-R recognition and cleavage motif, conjugated with FITC and biotin at their N and C-terminus, respectively (WT: FITC-SK**H**SGHLQRQKRDW-K-biotin, Mutant: FITC-SK**Y**SGHLQRQKRDW-K-biotin; Fig. 2c). Following digestion by furin (Fig. 2d)[14], peptides were cleaved into two fragments based on the recognition preference of the protease (Fig. S1). Liquid chromatography-mass spectrometry (LC-MS)

analysis demonstrated that proteolytic cleavage of the mutated peptide was substantially decreased compared with that of the WT peptide (Fig. 2e). Thus, our data support the postulation that replacing histidine with tyrosine debilitates the anchoring of the peptide within the catalytic pocket of furin protease and putatively impairs N-cadherin protein maturation.

**CRISPR/Cas9-mutated knock-in mice**. As the human CDH2 shares a high degree of similarity with its murine ortholog (Fig. 1d), with notable evolutionary conservation in the vicinity of the consensus prodomain cleavage motif, we generated two founder lines of CRISPR/Cas9-mutated mice harboring the specific human p.H150Y substitution in the *Cdh2* mouse ortholog; Selected KI F0 mice were bred with C57BL/6JRcc WT mice for two cycles to generate non-chimeric F1 KI heterozygotes. Heterozygote F1 offspring were then bred and F2 offspring of WT and mutant origin (*Cdh2^{H150T}* and *Cdh2^{H150Y(2)}* founder lines) were used for all further experiments (details in Methods). Thorough studies of the *Cdh2* homozygous knock-in (KI) 10-week-old mutant mice, with a specific focus on the brain, demonstrated no anatomical or histological abnormalities Fig. S2).

**Behavioral and cognitive phenotypes in *Cdh2*-mutant mice**. To assess possible in-vivo effects of the *Cdh2* mutation, we performed behavioral studies of the mutant mice. 10-week-old WT and homozygous *Cdh2^{H150Y}* KI mutant male mice ($n = 18$, nine mice per group) underwent a 3-week cassette of extensive phenotypic assessment (see Methods). *Cdh2^{H150Y}* mice exhibited significantly greater traveling distance, increased velocity, and prolonged mobility time in the open-field exploratory test (OFT), recapitulating motor-associated features of hyperactivity (Fig. 3a-f). No differences were observed when examining the duration mice spent in the center of the open-field arena, a measure inversely related to neophobic behavior, nor in the amount of center/border crossings. In addition, no difference was evident in the rotarod test (Fig. 3m), implying no significant effect on primary motor control. Regarding cognitive evaluation, a trend was observed in the spontaneous alteration test (Y-maze, Fig. 3h), suggesting a potential effect on executive functions and working memory. A significant difference was observed in the acoustic startle reflex test (ASR, Fig. 3j, k), demonstrating an elevated startle amplitude of the *Cdh2^{H150Y(1)}* mice, commonly associated with differences in sensorimotor integration. A similar pattern was observed in the pre-pulse inhibition test (Fig. 3l), in agreement with deficits in early-stage information processing, although no difference was evident in the pre-pulse attenuation fraction itself. Also, as part of the anxiety-domain assessment, no difference was demonstrated in the elevated plus-maze test (Fig. 3g). Lastly, the results of social interaction tests were inconclusive; although a significantly shorter interaction time in the resident-intruder test may imply a propensity for aggression (Fig. 3i), no differences were observed in the three-chamber sociability test followed by the social novelty test (Fig. 3n).

To further establish and validate the *Cdh2^{H150Y}* mouse model, we performed the OFT on a second founder line of homozygous mutant KI mice, denoted *Cdh2^{H150Y(2)}*. Congruent with the *Cdh2^{H150Y(1)}* results, 12-week-old male *Cdh2^{H150Y(2)}* mice ($n = 30$, 15 mice per group) exhibited a significantly greater traveling distance, increased velocity, prolonged mobility time, and a significant increase in the number of center zone alternation (Fig. 4 a-g) compared to C57BL/6JRcc control mice.

**Acute methylphenidate intervention aggravates locomotor activity in *Cdh2*-mutant mice**. Methylphenidate (MPH;

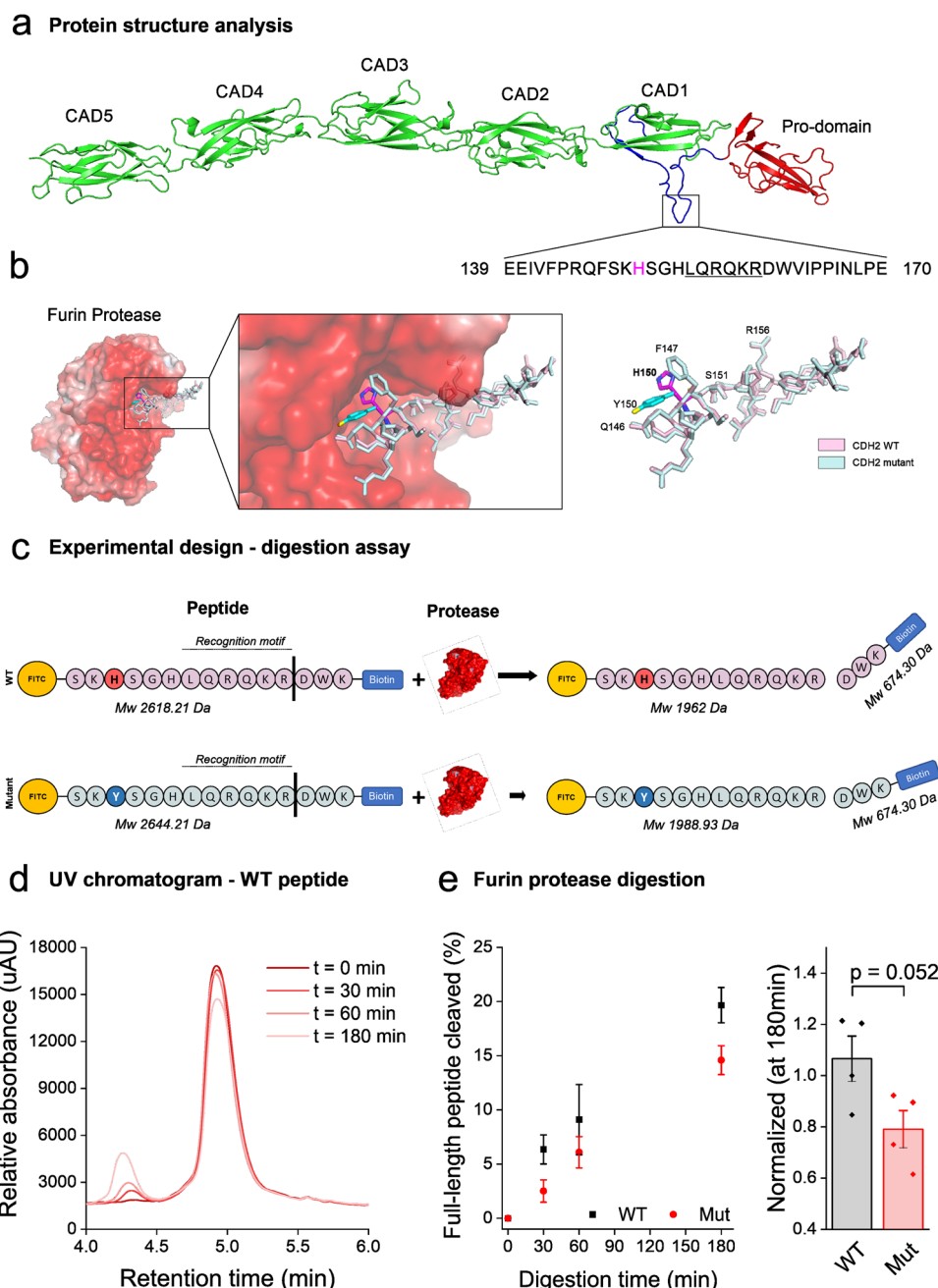

**Fig. 2 Structural analysis and disrupted cleavage of CDH2-mutated peptides. a** In-silico protein modeling. Ribbon representation of N-cadherin extracellular domains allows assessing the location of the identified mutation. Red, prodomain; green, extracellular cadherin domains (CADs 1-5); Blue, unstructured linker. The identified p.H150Y mutation resides within the unstructured region. Modeling was predicted using the SWISS-MODEL server (https://swissmodel.expasy.org/) based on the crystal structure of protocadherin GAmmaB4 extracellular domain (PDB ID 6E6B). **b** Electrostatic density representation map of furin protease (PDB ID 4Z2A). Furin cleavage site harbors the recognition motif RXK/R-R, which resides in proximity to the identified p.H150Y mutation site. Modeling was done using the HPEPDOCK server (https://omictools.com/hpepdock-tool), predicting protein-peptide interactions. The WT and CDH2-mutated sequences are denoted in pink and light blue, respectively; WT p.H150 residue is denoted with magenta, mutant p.Y150 residue with cyan. **c** Illustration of furin protease digestion assay; WT and mutant 22 amino-acid peptides, harboring the RXK/R-R recognition motif, were synthesized. Both peptides (10 μg) were digested by 2U enzyme furin protease followed by 30 °C incubation. Reaction mixtures were deactivated and subjected to LC-MS analysis. Predicted molecular weight is shown beneath each sequence. **d** Chromatogram separation and detection. Decreased absorbance of the digested full-length WT peptide at four-time intervals (0, 30, 60, 180 min) with a concurrent increase in absorbance of a fragmented small peptide (same was done for the mutant peptide). Peptides were separated by LC with subsequent tandem MS analysis. X axis: retention time (min), Y axis: relative absorbance (uAU). **e** Normalized cleavage efficacy after 180 min. The full-length peptide area under the curve (AUC) from MS analysis was calculated based on the desired spectral match. The ratio between AUC over time and the initial AUC at $t=0$ was extrapolated to calculate cleavage percentage. Left: percentage of peptide cleaved at $t=30$, 60, and 180 min. ($n=3$ for $t=30$ and 60, $n=4$ for $t=180$; $n$ represents an independent peptide digestion experiment subjected to LC-MS analysis (see Methods); mean ± SEM data were acquired, two-sided Student's $t$-test, at $t=30$ ns $p=0.08$; $t=60$ ns $p=0.44$; $t=180$ ns $p=0.052$). Right: normalized cleavage efficacy at $t=180$. Cleavage disruption of the mutated peptide is demonstrated with digestion >20% weaker in comparison with the WT peptide. UV, Ultraviolet; LC, liquid chromatography; MS, mass spectrometry.

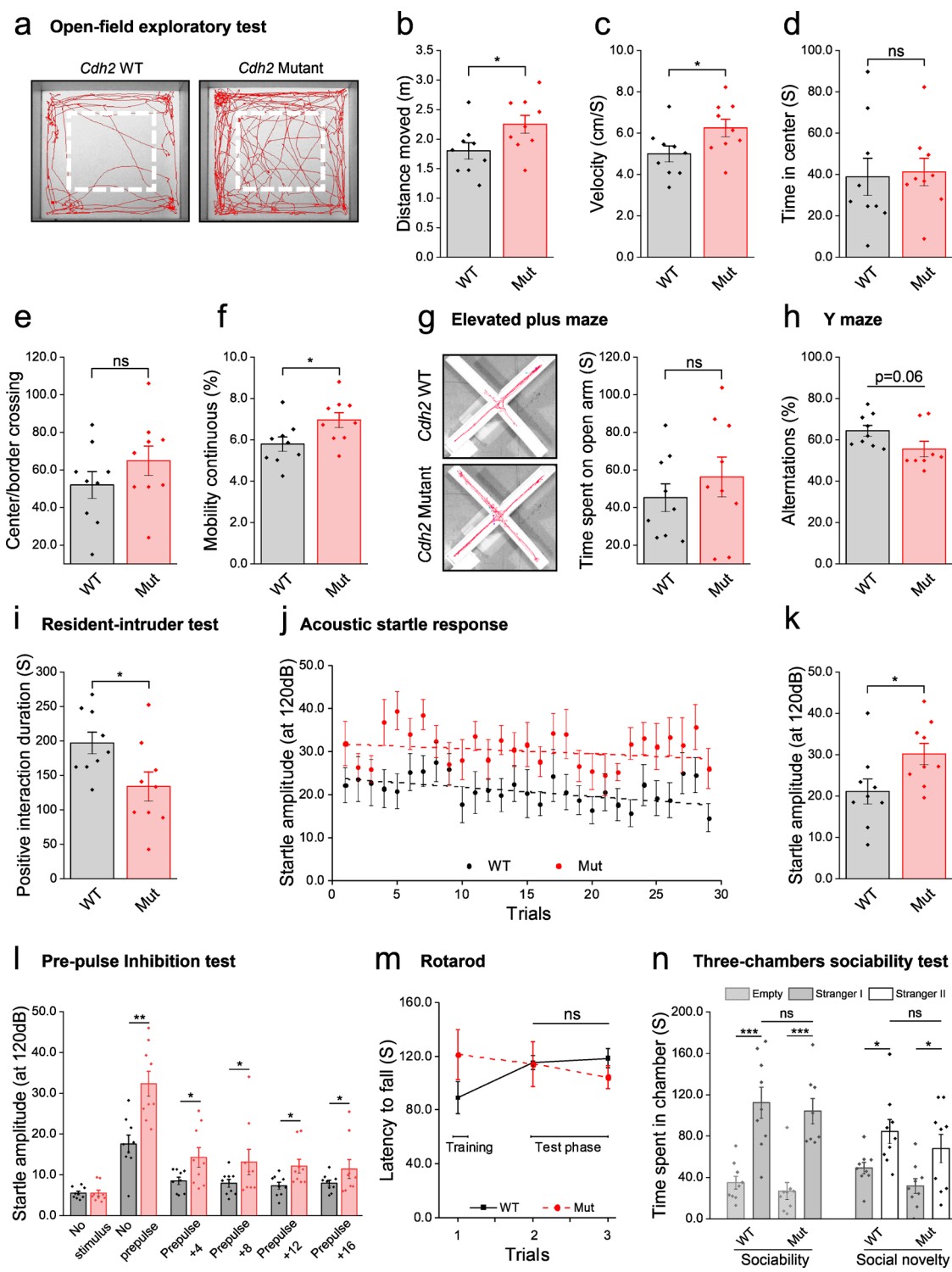

marketed under trade name Ritalin®, Methylin® or Concerta®), a schedule II CNS stimulant that increases dopamine tone, is one of the most commonly prescribed stimulants medications for symptomatic management of ADHD[15]. While long-term MPH administration is known to have a therapeutic effect, in acute dosing it has been shown to have anxiogenic effects[16]. Hence, to further examine in-vivo ramifications of the *Cdh2* mutation, we investigated MPH's effect on explorative behavior and sensorimotor gating through OFT and ASR, respectively. Male 14-week-old C57BL/6JRcc WT and homozygous *Cdh2*^H150Y(1/2) mice (*n* = 39, Fig. 4h) were injected intraperitoneally with MPH or

vehicle (0.9% saline) at 10 mg/kg, 30 min before initiation of tests. While both WT and mutant mice exhibited increased locomotor activity following MPH administration, the effect was significantly greater in the *Cdh2* mutant mice (Fig. 4i). In addition, the fold-change of the cumulative duration within center zone of the mutant mice was significantly smaller compared with the WT mice. These results highlight the in-vivo pathogenicity of the *Cdh2* mutation, with the mutant mice being significantly more susceptible to stimulatory intervention following MPH administration. This was further indicated by the ASR results (Fig. 4j), as the MPH-treated KI mice demonstrated a significantly lower

**Fig. 3 Behavioral experiments in Cdh2 knock-in mice.** Behavioral test results of $Cdh2^{H150Y(1)}$ and WT mice following cassette of motor, anxiety, cognitive and social interactions domains assessment. **a–f** Exploratory OFT consists of a square arena measuring 50X50X33cm. Mice explored the arena for 6 min, while their location was recorded. **a** OFT tracking visualization. $Cdh2^{H150Y}$ mice exhibited (**b**) significantly greater traveling distance (*$p = 0.04$), (**c**) increased velocity (*$p = 0.04$) and (**f**) prolonged mobility time (*$p = 0.03$). **d**, **e** No differences were observed examining the duration mice spent in the center of the arena nor at the amount of center/border crossings. **g** Elevated plus-maze test consisting of two sets of opposing arms extending from a central platform. One set of arms was enclosed by a 15 cm wall, while the other was open. No differences were observed examining the duration mice spent in each set of arms (ns $p = 0.4$). **h** Spontaneous alteration test (Y-maze) consisting of three arms in $120^0$ Y-shaped maze. An alteration was defined as a complete cycle of consecutive entrances into each of the three arms. WT mice exhibited more alterations compared to $Cdh2^{H150Y(1)}$ mice (ns $p = 0.06$). **i** Resident-intruder test assesses the time for aggressive social interaction. $Cdh2^{H150Y(1)}$ mice demonstrated significantly shorter interaction time (*$p = 0.03$). **j–l** ASR test consists of a single noise burst (120 dB, 40 ms); thereafter, the amplitude of the animal flinch is recorded. $Cdh2^{H150Y}$ mice demonstrated a significantly elevated startle amplitude (*$p = 0.03$). This pattern was consistent with the pre-pulse inhibition test. **m** Rotarod test, assessing motor abilities, demonstrated no difference between the groups. **n** No differences were observed in the three-chamber sociability test, followed by a social novelty test. For all experiments, mean ± SEM data were acquired; $n = 18$, 9 mice per group, two-sided Student's $t$-test. OFT, open-field test; ASR, acoustic startle reflex.

startle amplitude compared with their WT counterparts, as opposed to the non-MPH ASR experiment (Fig. 3j).

**The Cdh2 mutation decreases the size of the presynaptic vesicle cluster.** In accordance with the well-studied role of N-cadherin in synaptic development[17] and the known deleterious effect that aberrant N-cadherin prodomain cleavage has on synaptogenesis[18,19], we sought to investigate possible effects the mutation might have on synaptic properties in ex-vivo hippocampal neurons of $Cdh2^{H150Y}$ homozygous mice as compared to their WT counterparts. First, we quantified the total vesicle population by semi-quantitative immunofluorescence (IF) analysis of synaptobrevin2 (Syb2, an integral synaptic vesicle (SV) SNARE protein). Analysis was performed at two time points: 8 days in-vitro (DIV) and 14 DIV, which represent immature and mature synapses, respectively[20]. The IF intensity of the SV cluster in $Cdh2^{H150Y}$ neurons was significantly lower at 8 DIV, and more prominently so at 14 DIV (Fig. 5a, b). The decrease in the IF signal was accompanied by a sharper longitudinal distribution of SVs along the axis of the presynaptic terminal. This was quantified by determining the Full Width at Half Maximum (FWHM) of Gaussian fits to linear profiles of syb2 IF (Fig. 5c). These results imply that SV clusters in $Cdh2^{H150Y}$ synapses are smaller and more compact. We repeated this experiment performing IF for vGlut1, a vesicular glutamate transporter which partakes in a different synaptic function than sy2b. Although no difference was observed in 8 DIV neurons, 14 DIV neurons exhibited a significantly lower vGlut1 peak signal and a narrower distribution, similar to Syb2 at DIV 14 (Fig. 5a, b). Thus, the SV cluster in presynaptic terminals of $Cdh2^{H150Y}$ neurons is smaller.

**The readily releasable pool size is smaller in Cdh2-mutated neurons.** The SVs in the synapse can be functionally categorized into different subpopulations based on their availability for release. The recycling pool (RcP) contains SVs that can engage in stimulus-evoked synaptic release, while the resting pool (RtP) vesicles are not released during stimulation but can contribute SVs to the RcP for future rounds of release[21–23]. The readily releasable pool (RRP), a subgroup of the recycling pool located directly at the active zone[24], contains the vesicles that are the first to undergo fusion during synaptic activity, meaning they have the highest fusion probability[25]. Finally, the relative division of SVs between these pools can be modulated[26,27], thus altering the neuronal synaptic properties. Using the styryl-dye FM1-43, we stained the RcP SVs[28] and assessed the relative size of the RRP and the RcP. To quantify the RRP, we measured FM1-43 destaining during a brief burst of stimulation (20 stimuli at 20 Hz)[29] and found it to be significantly smaller in $Cdh2^{H150Y}$ neurons, while no difference was observed during prolonged FM1-43 destaining (Fig. 5d). These results suggest that while

the RcP is unaffected (see also Fig. 6c), the RRP is relatively smaller in the $Cdh2^{H150Y}$ neurons, consistent with weaker synaptic transmission.

**Synaptic release is attenuated in Cdh2-mutated neurons.** The size of the RRP correlates with the synaptic release probability which is related to synaptic strength[22,30]. To monitor synaptic function, primary neuronal hippocampal cultures from WT and $Cdh2^{H150Y}$ homozygous littermates were infected to express hSyn:synaptopHluorinX2 (sypHy), a genetically-encoded synaptic release sensor. The fluorescence of sypHy is quenched by the acidic lumen of intact synaptic vesicles. During SV exocytosis, fluorescence increases but decays back to baseline during recycling, after SV endocytosis and reacidification[31]. $Cdh2^{H150Y}$ neurons exhibited a lower sypHy signal upon electrical stimulation (20 Hz, 15 s) compared to WT neurons ($\Delta F/F_0$, Fig. 6a, b), illustrating weaker synaptic vesicle recycling in the mutant neurons. This result implies a decrease in the number of SVs released during stimulation of $Cdh2^{H150Y}$ neurons. To test whether this is only due to a decrease in the total number of SVs in $Cdh2^{H150Y}$ terminals, or also due to a decrease in the relative size of the RcP out of the total pool of SVs, we employed the alkaline trapping method (Fig. 6c)[32]; when sypHy is imaged in the presence of bafilomycin A, a blocker of the SV proton pump, sypHy reports only the cumulative exocytotic portion of the vesicle cycle; the decrease in sypHy fluorescence that is induced by compensatory endocytosis and reacidification is blocked, without affecting the kinetics of recycling[33]. Perfusion with saline containing $NH_4Cl$ at the conclusion of the experiment unquenches the intact SVs, thus revealing the entire vesicle population; the ratio between the plateau observed during exhaustive stimulation and the total population allows calculation of the relative size of the RcP. We observed no difference in the RcP fraction in WT and $Cdh2^{H150Y}$ synapses, which was approximately 50% (Fig. 6c). Considering $Cdh2^{H150Y}$ neurons exhibit smaller SV clusters, we conclude that their RcP are proportionally smaller and thus exhibit weaker synaptic release.

**Lack of effect of the Cdh2 mutation on the kinetics of vesicle recycling.** To assess whether the $Cdh2$ mutation affects synaptic vesicle recycling kinetics, we sought to measure the kinetics of the sypHy signal rising phase by supplementing bafilomycin A, where we could quantify the kinetics of exocytosis during stimulation trains. No difference was observed in the time constant of the signal increment (Fig. 6c), suggesting the properties of SV usage and recruitment are not affected. To examine a possible effect on the kinetics of endocytosis, we calculated the time constant of the decay of the SypHy signal after the termination of stimulation, in experiments performed in the absence of bafilomycin A. During this phase, no exocytosis is evoked, and only compensatory

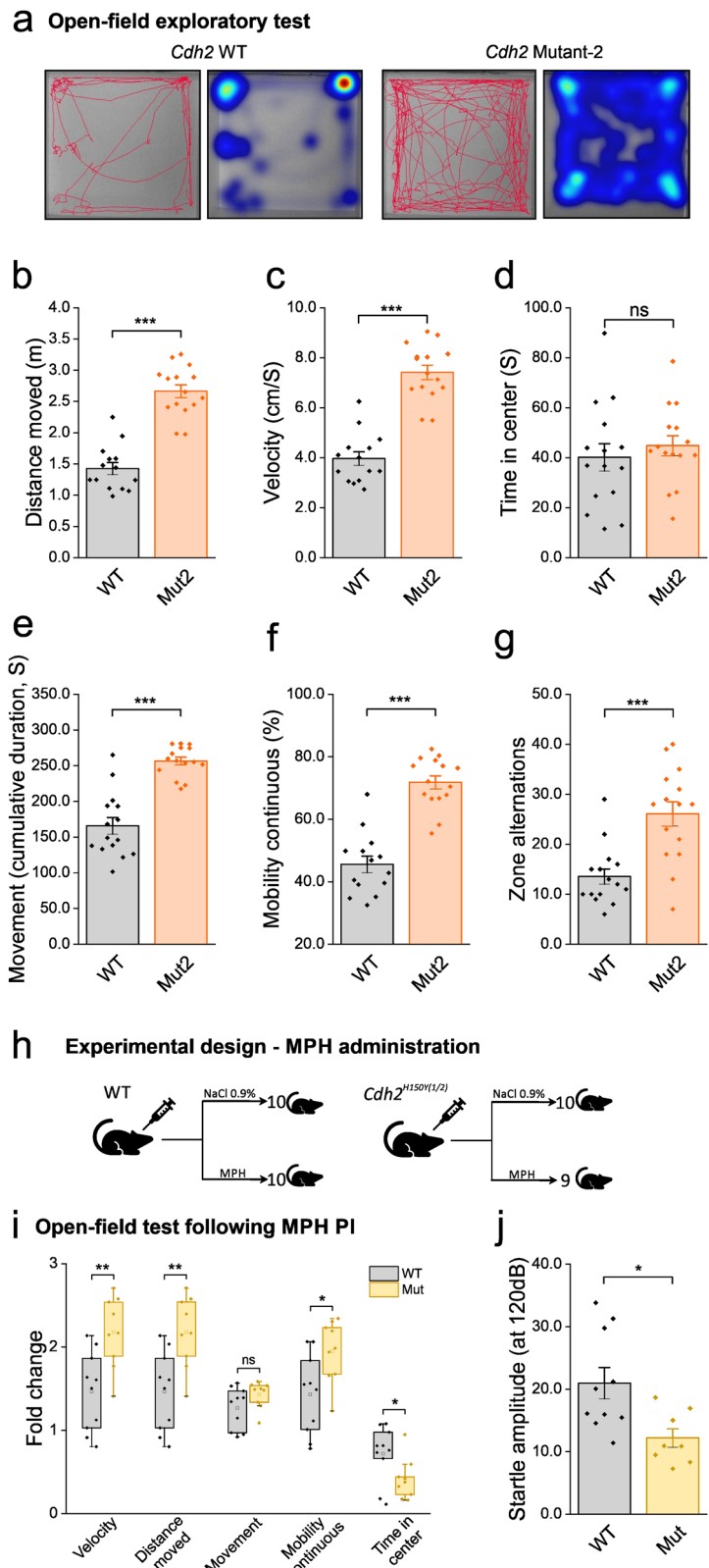

**a  Open-field exploratory test**

*Cdh2* WT

*Cdh2* Mutant-2

**h  Experimental design - MPH administration**

**i  Open-field test following MPH PI**

endocytosis takes place (Fig. 6a, upper inset). No difference was observed between the two genotypes, further demonstrating that the mutation had no impact on the kinetics of SV recycling.

**The frequency of spontaneous synaptic release is lower in *Cdh2* slices**. A reduction in total vesicle population or SV clusters in presynaptic terminals and a decrease in the RRP could lead to a

decrease in the frequency of spontaneous synaptic release[30,34,35]. Thus, we performed patch-clamp recordings from pyramidal neurons in the CA1 in the hippocampus in acute brain slices from WT and *Cdh2^{H150Y}* littermates and recorded spontaneous miniature excitatory postsynaptic currents (mEPSCs) in the presence of Tetrodotoxin (TTX, 1 nM) and Bicuculline (GABA A antagonist, 10 μM) (Fig. 6d-f). The mEPSC frequency was

**Fig. 4 Further behavioral experiments and methylphenidate intervention. a–g** Exploratory OFT of $Cdh2^{H150Y(2)}$ (second founder line). **a** OFT tracking visualization. Congruent with $Cdh2^{H150Y(1)}$ results, $Cdh2^{H150Y(2)}$ mice further exhibited (**b**) significantly greater traveling distance (***$p = 2E^{-9}$), (**c**) increased velocity (***$p = 2E^{-9}$), **e, f** prolonged mobility and cumulative movement duration (***$p = 1E^{-7}$, $1E^{-8}$ respectively) and (**g**) significant increase in the number of center/border crossings (***$p = 1E^{-4}$). **d** No difference was observed examining the duration mice spent in the center of the arena. For all OFT results, mean ± SEM data were acquired ($n = 30$, 15 mice per group, two-sided Student's $t$-test). **h** Experimental design illustrating MPH administration. Male 14-week-old C57BL/6JRcc WT and $Cdh2^{H150Y(1/2)}$ mice were injected intraperitoneally with MPH or vehicle (0.9% saline solution) at 10 mg/kg, 30 min before initiation of tests. **i** Five domains of an OFT following MPH administration are presented as fold-change relative to vehicle-treated controls. While both WT and mutant mice exhibited increased locomotor activity, the fold-change in velocity and distance traveled (**$p = 0.0032$) and mobility continuous (*$p = 0.0174$) domains was significantly greater in the $Cdh2$ mice. To note, the fold-change of the cumulative duration within center zone of the mutant mice was significantly smaller compared with the WT mice (*$p = 0.034$). Symbols represent single animal; bars, 10–90% percentiles; box, 25–75% percentiles ($n = 39$, 10 mice per group, 9 mice in the MPH-treated mutants, two-sided Student's $t$-test, ns $p = 0.117$). **j** ASR test results of MPH-treated mice. $Cdh2^{H150Y(1/2)}$ mice demonstrate significantly lower startle amplitude following acute MPH administration compared with WT controls. Mean±SEM are represented ($n = 18$, 10 WT and 8 mutant mice, two-sided Student's $t$-test, *$p = 0.012$). Orange bars represent the second founder line, yellow bars represent an experiment with both $Cdh2^{H150Y1/2}$ founder lines. OFT, open-field test; PI, peritoneal injection; MPH, methylphenidate; ASR, acoustic startle reflex.

significantly smaller in $Cdh2^{H150Y}$ neurons; in contrast, no difference was observed in the mEPSC amplitudes. The latter (also termed the quantal size) is generally associated with changes in postsynaptic properties such as receptor density[36], while the former is typically attributed to changes in presynaptic properties – either the availability of vesicles or the per-vesicle probability of spontaneous release. These results strengthen our previous observation that the SV clusters in presynaptic terminals of $Cdh2^{H150Y}$ neurons are smaller as are their RRPs.

**Schaffer collateral pathway synapses from $Cdh2$-mice exhibit lower frequency facilitation.** Short-term synaptic plasticity (STP) regulates the activity of neural networks and information processing in the nervous system on a timescale of milliseconds to minutes. One form of STP is synaptic facilitation, a phenomenon in which postsynaptic potentials are augmented during short bursts of repetitive stimulation. Multiple presynaptic mechanisms have been suggested to account for facilitation, including accumulation of calcium in the terminal during the stimulation train that leads to a progressive increase in the probability of release ($P_r$)[37–39]; but facilitation was also correlated with the size of the RRP[30,34]. To examine STP, we compared the change in excitatory postsynaptic potential responses (fEPSPs) in extracellular recordings from the CA1 area in acute brain slices of $Cdh2^{H150Y}$ and WT mice during the delivery of 5 stimuli to the Schaffer collaterals at different frequencies (Fig. 6g-i). At higher stimulation frequencies, $Cdh2^{H150Y}$ slices exhibited lower frequency facilitation than WT slices, consistent with the smaller presynaptic cluster in their neurons. Using synaptophysin 1-oGCaMP6f, in which the calcium indicator GCaMP6f is fused to the SV protein synaptophysin I, we examined baseline calcium and its accumulation in the terminals during field stimulation. No difference was observed between $Cdh2^{H150Y}$ and WT neurons (Fig. 6j-l). While this result implies that calcium-handling is not different in neurons from the two genotypes, it is of note that facilitation depends on calcium dynamics in nanodomains[37], which are not directly measurable using calcium imaging. Altogether, our results support a role for N-cadherin in modulating short-term synaptic plasticity, perhaps due to a decrease in the RRP size.

**$Cdh2$ mutation alters Tyrosine hydroxylase levels.** Impaired connectivity, mainly within the dopaminergic (DA) mesocortical pathway, has been implicated in the pathophysiology of ADHD. This circuit includes DA-releasing neurons at the ventral tegmental area (VTA), projecting to the prefrontal cortex (PFC). Notably, it has been demonstrated that N-cadherin regulates the

proliferation and differentiation of ventral midbrain (vMB) dopaminergic progenitors[6]. Therefore, to examine whether the $Cdh2$ mutation impairs dopaminergic distribution, we performed real-time quantitative PCR (qPCR) analyses examining canonical dopaminergic markers. For that purpose, we micro-dissected PFC and vMB brain tissues from male 13-week-old WT and homozygous $Cdh2^{H150Y}$ mice ($n = 18$, nine mice per genotype). Transcript levels of tyrosine hydroxylase (TH), the first enzyme in the biosynthesis of catecholamines, were significantly decreased in the PFC of the $Cdh2^{H150Y}$ mice, with a similar trend observed within the vMB (Fig. 7a, b). In contrast, no significant differences were observed in dopamine transporter ($DAT1$) levels. Following the qPCR findings, we performed IF analysis of the vMB ($-2.92$ to $-3.4$ relative to bregma), quantifying the density of TH-positive neurons in both genotypes. $Cdh2^{H150Y}$ specimens demonstrated a significant reduction in the number of TH-positive cells (Fig. 7c-e), implying a potential decrease in dopaminergic tone in mutated tissues.

**Decreased dopamine concentration in $Cdh2$-mutated prefrontal cortex.** With these results, we went on to test whether the observed changes in TH expression could affect dopamine concentration in mesocortical regions: we quantified dopamine levels using enzyme-linked immunosorbent assay (ELISA) in the PFC and vMB of male 14-week-old WT and homozygous $Cdh2^{H150Y}$ mice ($n = 20$, ten mice per genotype). In line with altered TH expression levels, we found dopamine concentrations to be significantly decreased in the PFC of the $Cdh2^{H150Y}$ mice (Fig. 7f, g). This suggests that dopamine tone in the mesocortical pathway is affected by the N-cadherin mutation.

**Whole-transcriptome analysis of $Cdh2$-mutated brain tissues.** To further decipher differentially expressed genes (DEGs) due to the $Cdh2^{H150Y}$ mutation, we performed transcriptome analysis of micro-dissected vMB and PFC brain tissues of 13-week-old male WT and homozygous $Cdh2^{H150Y}$ mice ($n = 16$, four samples per tissue per genotype). After applying adjusted $P$-values to the RNA-seq data, hierarchical clustering (Fig. 8a, b) identified 181 DEGs within the PFC, of which 99 were downregulated and 82 were upregulated in the mutants. Notably, $Cdh2$ transcripts were downregulated ($p = 2.7E^{-04}$, FC = $-1.49$), strengthening the pathogenicity of the mutation. Hierarchical clustering of the vMB identified 604 DEGs, with 383 and 221 genes down and upregulated in the mutants, respectively (Supplementary Data 1). Several intriguing genes associating with $Cdh2$ were differentially expressed in the mutants; $CTNND1$ ($p = 6.8E^{-03}$, FC = $-1.31$), coding for p120-catenin, was downregulated. P120 binds to the

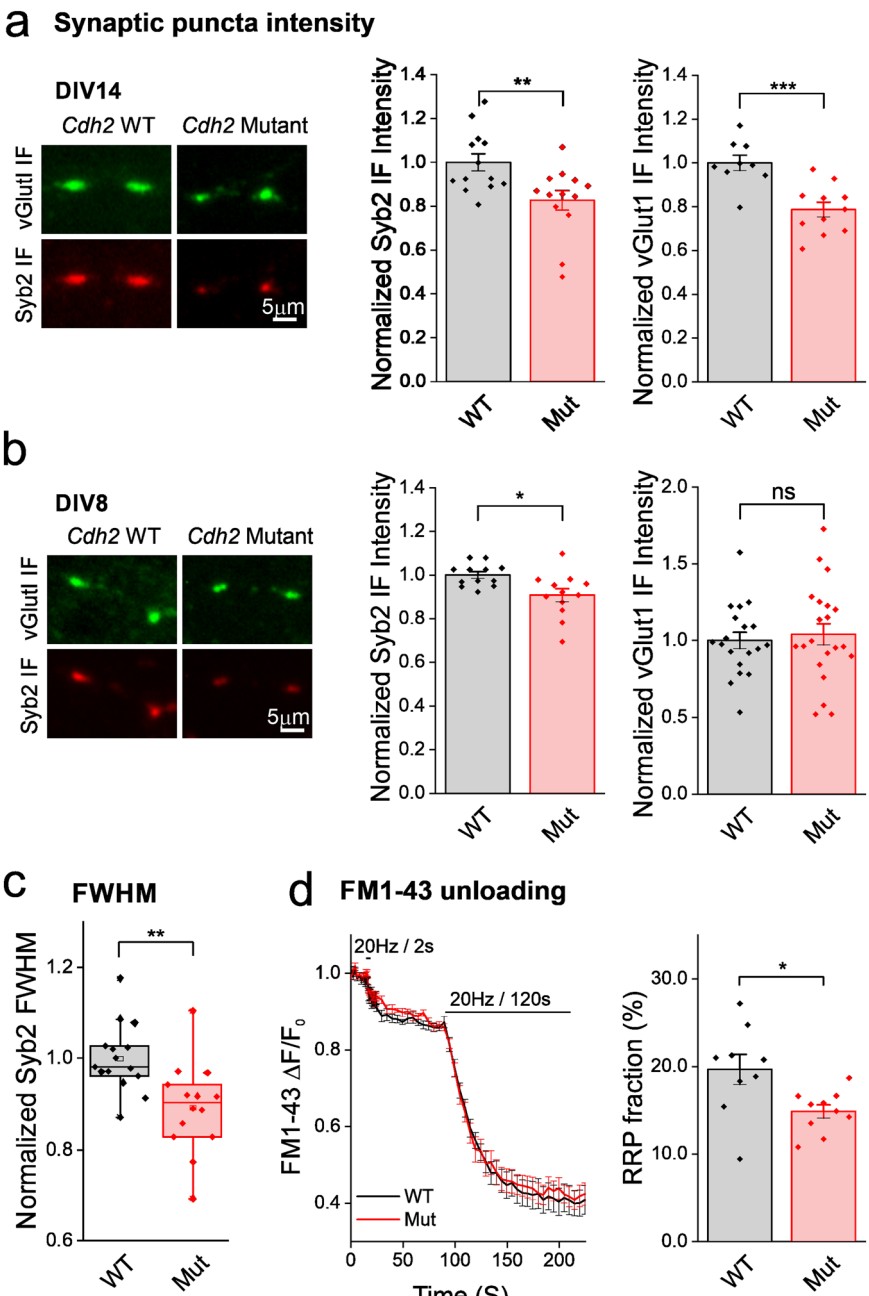

**Fig. 5 Cdh2 mutation decreases synaptic vesicle cluster size in presynaptic terminals. a** Left: hippocampal neurons from WT and *Cdh2^{H150Y}* mice were immunostained at 14 DIV for Syb2 (red) and vGlut1 (green). Right: mean ± SEM density of synaptic vesicles (SVs) within presynaptic terminals. Only synapses negative for GAD6 were included (Syb2: $n = 13$ images; vGlut1: $n = 9$ and 11 images, respectively, >500 boutons analyzed per image, two-sided Student's *t*-test, **$p = 0.007$, ***$p = 3E^{-4}$). **b** Same as in A for 8 DIV (Syb2: $n = 12$ and 13; vGlut1: $n = 19$ and 21 images, respectively, >500 boutons analyzed per image, two-sided Student's *t*-test, *$p = 0.01$, ns $p = 0.65$). **c** Quantifying the distribution of the Syb2 fluorescence in the synaptic puncta at rest by measuring their FWHM. Symbols represent single image FWHM; bars, 10–90% percentiles; box, 25–75% percentiles; middle bar, median ($n = 15$ and 14 images, respectively, >30 boutons analyzed per image, two-sided Student's *t*-test, **$p = 0.002$). **d** Progressive unloading of FM1-43 dye by electrical stimulation delivered at 20 Hz for 2 s, followed by a minute of recovery and then stimulated again for 120 s at 20 Hz. Left: mean ± SEM $\Delta F/F_0$ traces. Right: quantification of the readily releasable pool (RRP) fraction (percentage from the total release, $n = 9$ and 10 experiments, respectively, >30 boutons analyzed in each experiment, two-sided Student's *t*-test, *$p = 0.017$). DIV, days in-vitro; FWHM, Full Width at Half Maximum.

intracellular domain of N-cadherin and regulates excessive activation of the Wnt signaling pathway. P120-catenin is known to interact with *IQGAP1*, which was also downregulated ($p = 5.5E^{-04}$, FC = −1.39). Interestingly, N-cadherin is known to interact with *IQGAP1* in synaptic plasticity and remodeling. Another differentially expressed gene was *DDC* ($p = 4.8E^{-02}$, FC = +1.63), coding the DOPA carboxylase enzyme that catalyzes the decarboxylation of

L-3,4-dihydroxyphenylalanine (DOPA) to dopamine. *DDC* was upregulated, implying a potential compensatory mechanism. Further gene ontology (GO) analysis identified significantly enriched GO terms associating with biological processes highly relevant to behavioral features, including GABAergic and glutamatergic synapses, neuronal projection, synaptic organization, cell-cell adhesion, and axon guidance, terms which were previously linked

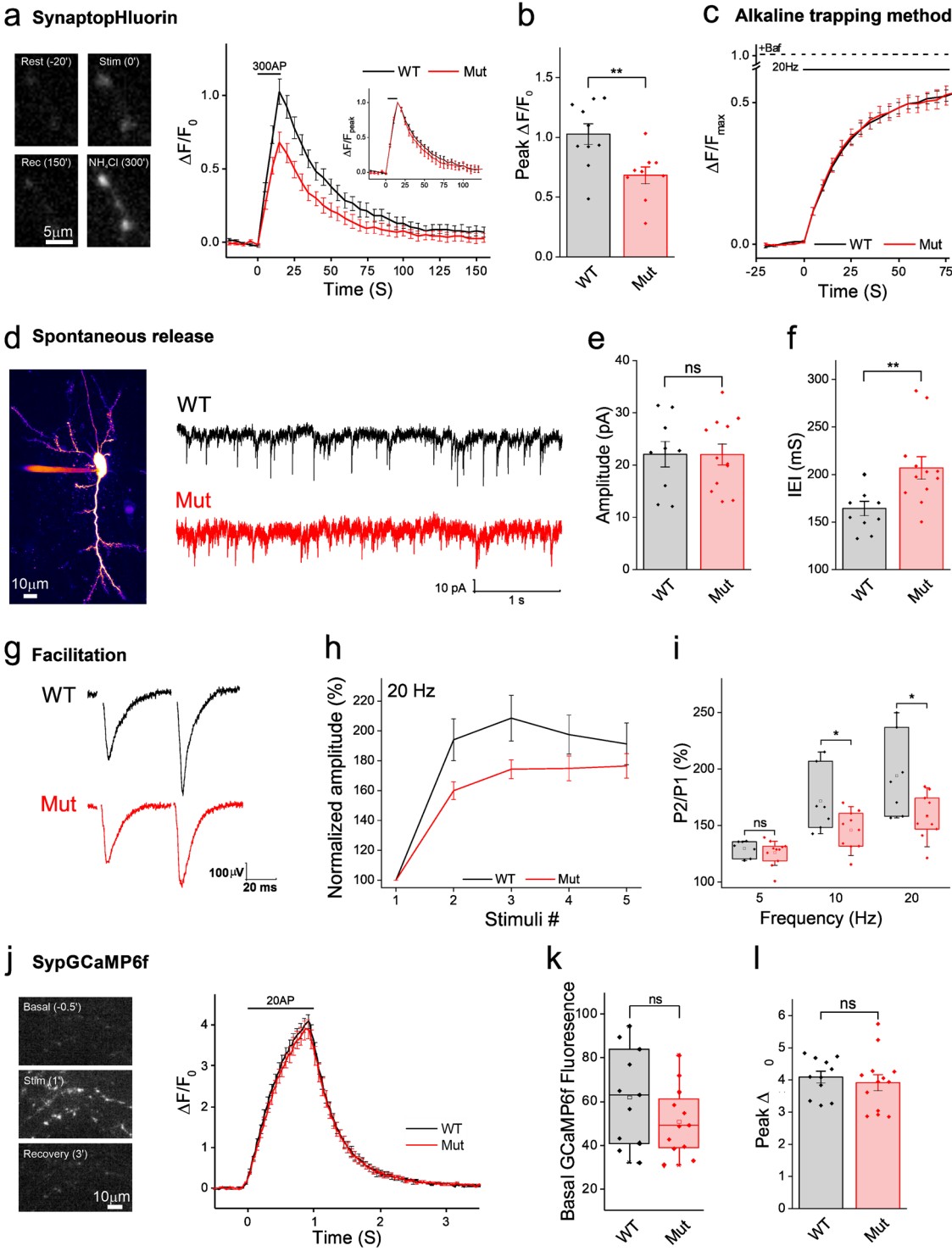

to aberrant N-cadherin functioning (Fig. 8c-e, Supplementary Tables 1-3). Importantly, a considerable number of phenotype-causing genes within these clusters have been previously linked with ADHD and were mostly downregulated in the mutants (Supplementary Table 4).

## Discussion

Cell adhesion molecules (CAMs) mediate cell-cell interactions in both the developing and mature nervous systems and are believed to play a critical role in cell migration, synapse formation, and target recognition. Deficiencies in these molecules were previously demonstrated to cause behavioral deviations related to neuropsychiatric disorders[40]. Among these CAMs, cadherins are a large family of calcium-dependent adhesion proteins, which constitute the principal adhesion elements mediating cell-cell cohesion[41]. Consistent with their expression in the nervous system, abnormalities in critical cadherin domains have been previously statistically linked through genome-wide association studies (GWAS) to psychiatric disorders, mainly autism spectrum disorder (*CDH9/10, CDH5, CDH11*), schizophrenia (*CDH8, CDH23, CDH12/18*), depression (*CDH13, CDH18, CDH28*) and bipolar disorder (*CDH7*)[42]. Notably, while *CDH13* (encoding

**Fig. 6 Synaptic release is attenuated in Cdh2-mutated neurons. a** Left: representative images of sypHy-puncta in WT neurons. Middle: mean ± SEM sypHy traces. Hippocampal neurons were field-stimulated (20 Hz, 15 sec) and the sypHy signal was monitored. Mean±SEM traces normalized to $F_{max}$ (at peak) are presented in the upper inset. **b** Peak mean ± SEM $\Delta F/F_0$ fluorescence obtained from traces in 6 A ($n = 9$ experiments, >60 boutons analyzed per image, two-sided Student's $t$-test, **$p = 0.007$). **c** Mean±SEM SypHy traces in neurons field-stimulated (20 Hz, 120 s) in the presence of Bafilomycin A. Graph shows the cumulative release of SVs, normalized by the total SV pool size (after addition of $NH_4Cl$). The plateau at the end of the stimulus indicates the recycling pool (RcP) relative size ($n = 9$ experiments, >60 boutons analyzed per image, two-sided Student's $t$-test, ns $p = 0.78$). **d** Intracellular recording of spontaneous mEPSCs. Left: reconstructed z-stack image. Right: representative traces recorded from WT (top) and mutant (bottom) slices under resting conditions, in the presence of TTX and bicuculline. Quantification of the mEPSC amplitude (**e**) and inter-event intervals (IEIs; **f**). Bars: mean ± SEM across recordings ($n = 10$ and 11 cells from 4 different mice, respectively, two-sided Student's $t$-test, ns $p = 0.99$, **$p = 0.0085$). **g** Representative fEPSPs were recorded from the CA1 area, delivering stimuli to the Schaffer collaterals at 20 Hz in acute slices prepared from WT and $Cdh2^{H150Y}$ mice. **h** fEPSP amplitudes normalized by the first response in each train (mean ± SEM), in WT and $Cdh2^{H150Y}$ slices at 20 Hz ($n = 7$ and 10 slice recordings acquired from 3 mice, respectively). **i** Quantification of the P2/P1 ratio at all frequencies. Symbols represent single slice recordings; bars, 10–90% percentiles; box, 25–75% percentiles ($n = 7$ and 10–11 slice recordings acquired from 3 mice, respectively, two-sided Student's $t$-test, 5 Hz: ns $p = 0.43$; 10 Hz: *$p = 0.032$; 20 Hz: *$p = 0.02$). **j** Neurons expressing the SypGCaMP6f construct were field-stimulated (20 Hz, 1 s). Left: representative images of WT neurons expressing SypGCaMP6f. Right: mean ± SEM SypGCaMP6f traces (**k**) Quantification of average basal fluorescence in 6 J. Symbols represent single image F0; bars, 10–90% percentiles; box, 25–75% percentiles; middle bar, median. **l** Quantification of peak $\Delta F/F_0$ in 6 J represented as mean ± SEM ($n = 11$ and 12 experiments, respectively, >50 boutons analyzed per image, basal: ns $p = 0.18$; peak $\Delta F/F_0$: two-sided Student's $t$-test, ns $p = 0.76$). mEPSCs, miniature excitatory postsynaptic potentials; fEPSP, field excitatory postsynaptic potential; TTX, Tetrodotoxin.

T-cadherin, a GPI-anchored protein with cell adhesion properties that is highly expressed in the brain and cardiovascular system) has been possibly associated with ADHD through GWAS with no causative or functional proof[43–45], to date cadherins have not been implicated in the pathophysiology of ADHD[46]. We here describe the first monogenic non-syndromic familial ADHD and delineate through human and mouse studies the role of aberrant *CDH2*-related pathways in ADHD pathophysiology.

*CDH2* encodes N-cadherin, a type I classical cadherin with an essential role in the early steps of brain morphogenesis[47]. N-cadherin is initially synthesized as an adhesively inactive precursor bearing a prodomain, thought to limit adhesion during the early stages of biosynthesis[48]. This prodomain processing is highly important in N-cadherin maturation, with aberrant cleavage impairing its adhesive properties, resulting in impaired synapse formation, mainly through enhancement in migratory properties. The p.H150Y mutation we identified is within the recognition domain of the protease responsible for this endogenous modification, and as we demonstrate through biochemical studies, impacts this tightly regulated post-processing. Also, long-term expression of an aberrant N-cadherin was demonstrated to reduce synapse connectivity[49]. Accordingly, our in-vitro results support deficits in synaptic formation as an intriguing mechanism underlying the patients' clinical manifestations.

N-cadherin has been further demonstrated to regulate presynaptic function[50] and to control presynaptic vesicle clustering through *trans*-synaptic mechanisms, promoting the accumulation of proteins involved in the synaptic organization. Importantly, in the absence of N-cadherin, synaptic vesicle clustering is impaired[51]. These observations are consistent with our results, exhibiting a significant decrease in presynaptic vesicle clustering in *Cdh2^{H150Y}* neurons, supporting our conclusion that the p.H150Y mutation impairs N-cadherin function. While N-cadherin deletion is embryonically lethal[52], in-vivo knockdown studies have shown to generate defects in synaptic homeostasis, between excitatory and inhibitory synaptic markers, the latter of which have been extensively studied in relation to ADHD[53]. In addition, a recent study demonstrated that cadherins, specifically N-cadherin, interact with synaptic organizers to promote synaptic differentiation[54]; using CRISPR-Cas9 mutated HEK293 cells, lacking the *Cdh2* gene, CDH2 was demonstrated to promote neurite branching, required for three synaptic organizers, neurologin1 (NLGL1), leucine-rich repeat transmembrane protein 2 (LRRtm2), and Cell Adhesion Molecule 1 (Cadm1/SynCAM) to stimulate specifically presynaptic differentiation.

The association between ADHD and altered modulation of synaptic transmission has been well-studied, mainly through the DA pathways, as psychostimulants that are used to treat ADHD target mostly DA transmission. As aforementioned, the VTA neurons project to several limbic structures, mainly the ventral striatum and the PFC, giving rise to both the mesolimbic and mesocortical pathways. The latter has been implicated as a critical circuit essential for motivated behaviors and other cognitive functions related to ADHD[55]. Interestingly, recent studies have demonstrated that loss of N-cadherin leads to a reduction in the number of DA neurons, mediated by the canonical Wnt/β-catenin signaling pathway. Furthermore, it was demonstrated that β-catenin, co-localized with N-cadherin, controls DA neurogenesis by maintaining the integrity of the vMB[56]. Our results support these findings, demonstrating a reduction in the density of TH-positive neurons within the PFC and vMB of the *Cdh2^{H150Y}* brains, with a concurrent decrease in dopamine concentration. Taken together, these findings delineate an association between a deficiency in adhesive properties of N-cadherin, impaired synaptic function, and potential decrease in dopaminergic tone, with the current notion of ADHD as a "hypo-dopaminergic" condition[57].

Concordant with previous studies[58], WT mice demonstrated increased locomotor activity following acute administration of MPH, the most commonly prescribed psychostimulant for ADHD patients; notably, this effect was greater in the *Cdh2^{H150Y}* mice, suggesting that the mutant mice are significantly inclined to stimulatory mediation, further indicating in-vivo pathogenicity of the *Cdh2^{H150Y}* mutation. Considering that significant portions of ADHD patients do not respond to amphetamines or MPH administration, *Cdh2^{H150Y}* mice could model human ADHD patients and serve in the discovery of novel treatment modalities for cases not responsive to amphetamines or MPH.

As mentioned, the Wnt/β-catenin pathway may be a potential mechanism by which aberrant N-cadherin adhesive properties might affect dopamine levels, as β-catenin constitutes a key component in the N-cadherin/β-catenin complex. Consistent with our findings of impaired presynaptic activity, it has been demonstrated that the N-cadherin/β-catenin complex has a principal role at nascent synapses in the assembly of presynaptic vesicle clusters, as the expression of non-functional N-cadherin greatly reduces the size and function of presynaptic terminals. Interestingly, our enrichment analysis delineated potential association between downregulated N-cadherin, P120-catenin, and IQGAP1, perhaps through the Wnt signaling cascade, which, among others, regulates presynaptic vesicle clustering.

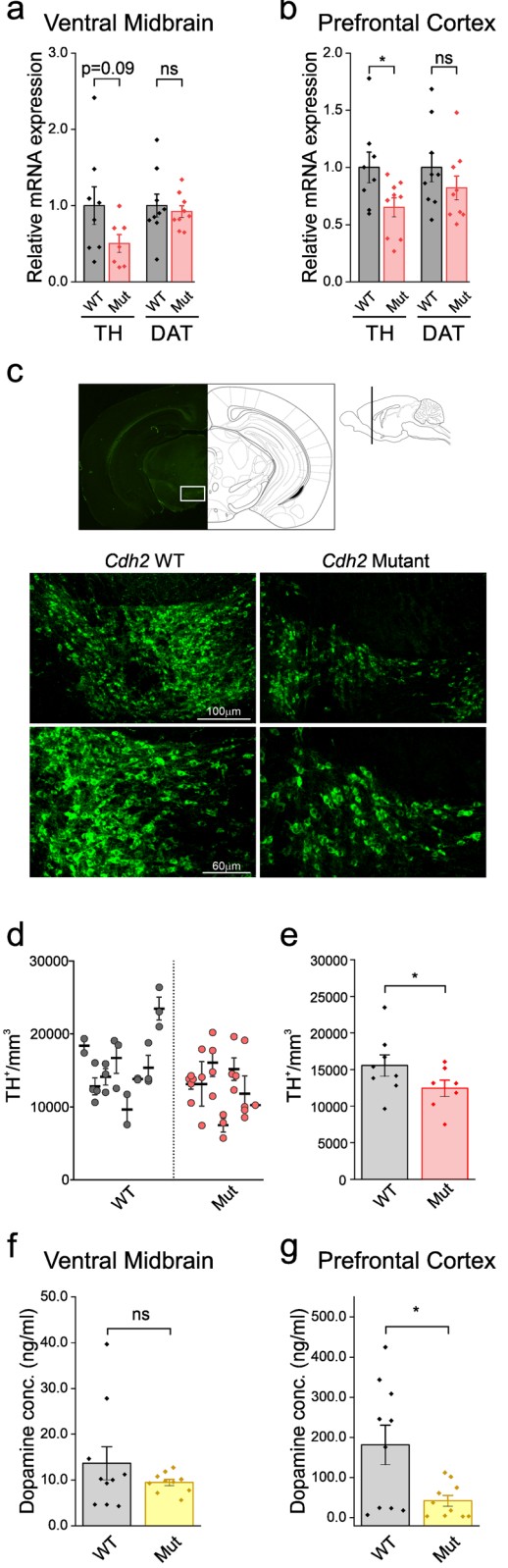

**Fig. 7 Cdh2 mutation alters Tyrosine hydroxylase levels and dopamine concentration. a** Relative TH and DAT mRNA expression levels. RNA was extracted from micro-dissected vMB tissue samples in both the WT and Cdh2$^{H150Y}$ 13-week-old mice. Mean±SEM data were acquired from at least three independent experiments) $n = 18$, two-sided Student's $t$-test, left: ns $p = 0.09$, right: ns $p = 0.64$. **b** Same as A in the PFC ($n = 18$, two-sided Student's $t$-test, left: *$p = 0.03$, Right: ns $p = 0.28$). **c** TH immunofluorescence visualization. WT and Cdh2$^{H150Y}$ hemispheres were dissected to 50-micron frozen floating sections, taken from −2.92 to −3.4 millimeters relative to bregma. **d, e** TH-positive neurons immunofluorescence analysis in the vMB. Five sections were analyzed for each mouse ($n = 18$). The number of TH-marked cells was manually counted at x10 magnification in a defined area containing the vMB. The number of visible TH-stained cells was divided by the area volume, calculated as follows: (area*number of stacks*stack spacing*µm/pixels$^2$)/10$^9$. **d** Each sub-column in the scatterplot depicts specimens obtained from an individual mouse, highlighting mean ± SEM. **e** Bar graph of the same data, depicting predicted marginal mean ± SEM averaged for each experimental group. Cdh2$^{H150Y}$ specimens demonstrated a significant reduction in the number of TH-positive cells. As directionality was pre-hypothesized based on earlier convergent data from RT-qPCR, a one-tailed 0.05 significance threshold was set (Significance based on mixed model followed by one-tailed distribution of z-score, *$p = 0.0435$). **f, g** Dopamine concentration levels. PFC and vMB tissue dopamine concentration is decreased with Cdh2$^{H150Y}$ specimens. Shown are the measured mean ± SEM dopamine levels using high sensitivity ELISA in (**f**) PFC ($n = 10$, two-sided Student's $t$-test, *$p = 0.013$) and (**g**) vMB ($n = 10$, two-sided Student's $t$-test, ns $p = 0.27$). Results were provided using 4 Parameter Logistic analyses, with all determinations performed in duplicates. TH, tyrosine hydroxylase, DAT dopamine transporter; vMB ventral midbrain, PFC prefrontal cortex.

to gate intrusive sensory, motor, and cognitive information[59]. In this regard, another aspect demonstrated in our work was a significant alteration in the amplitude of the acute ASR observed in the behavioral evaluation of the Cdh2$^{H150Y}$ mice as compared to WT controls, with or without stimulatory intervention. ASR disruptions have been well-documented in several psychiatric disorders, including ADHD, and are thought to result from impairment in sensorimotor gating, with several nuclei within the midbrain directly modulating intrinsic ASR circuits[60,61]. Supporting our results, previous studies have shown that disrupted sensorimotor gating may result from dysfunctional dopamine neurotransmission, involving abnormal activation of the Wnt system[62].

To date, despite its biological significance, only a limited number of publications described phenotypes associated with CDH2 variants; a notable study associated canine compulsive disorders with its CDH2 ortholog[63], whereas several SNPs have been associated in humans with obsessive-compulsive and Tourette disorders[64]. Recently, de novo heterozygous variants in CDH2 were described in nine unrelated individuals, presenting diverse clinical manifestations, whose main features were neurodevelopmental[65]. Pathologies described within this study were ocular (7/9), urogenital (4/9), cardiovascular (7/9), hearing loss (2/9), axial hypotonia (3/9), epilepsy (2/9), intellectual disability (6/9), and neuropsychiatric issues (7/9). Though carefully studied, none of our heterozygous individuals presented with any clinical abnormalities. Nonetheless, none of the nine heterozygous variants were found to reside in proximity to our mutation, with the majority affecting residues of the EC4 to EC5 linker region. Thus, one cannot rule out that haploinsufficiency mutations in different domains might cause different phenotypes affecting different tissues, likely reflecting, at least in part, tissue-specificities of adhesive interactions.

Nonetheless, further studies are required to decipher the exact mechanism by which these impairments occur.

Deficiencies in perceptual capacity in ADHD patients are thought to result in sensory overload, which may underlie symptoms of inability to regulate attention performance. The inability to filter out extraneous stimuli implicates a deficit in sensory gating, characterized by a general reduction in the ability

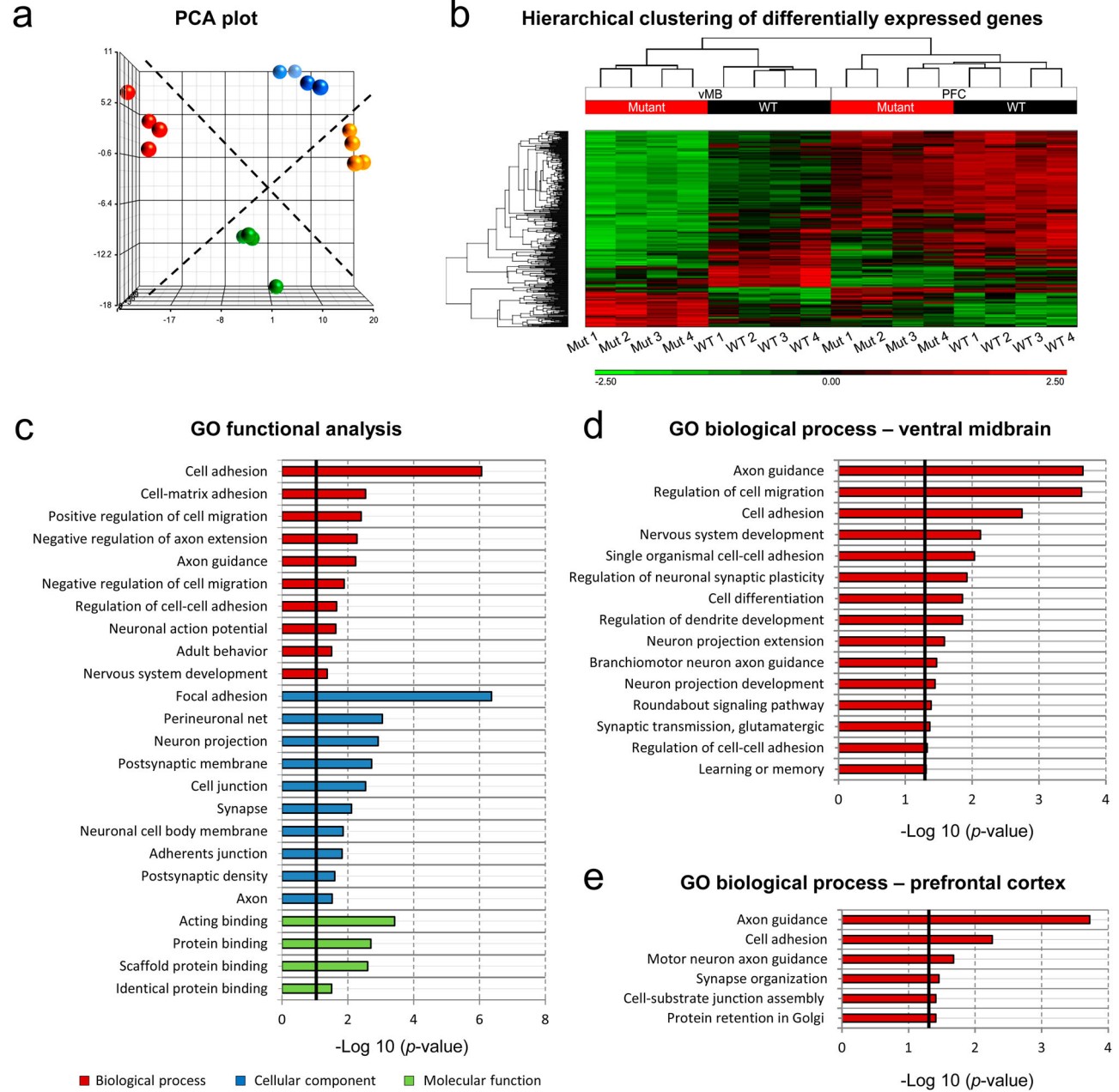

**Fig. 8 Whole-transcriptome analysis. a** PCA plot, demonstrating hierarchical clustering of WT and *Cdh2*$^{H150Y}$ mice ($n = 16$, four samples per tissue per genotype), extracted from both the PFC (blue and yellow spheres, respectively) and vMB (red and green spheres, respectively) tissue samples. **b** Heatmap of DEGs clustering for PFC and vMB RNA-seq data. **c** GO term enrichment analysis of 416 DEGs from vMB and PFC using the DAVID functional annotation tool (https://david.ncifcrf.gov/summary.jsp/). The DAVID Functional Annotation Clustering function uses a Kappa statistic score to measure relationships among the annotation terms based on the degrees of their co-association genes. The most significantly relevant GO terms in biological process (red), cellular component (blue), and molecular function (green) branches are presented. All statistically significant values of the terms were negative 10-base log-transformed. The vertical black line delineates the significance threshold (corresponding to *p*-value = 0.05). **d**, **e** GO analysis using DAVID functional annotation tool, presenting the most significant relevant terms in the biological process domain for (**d**) 604 vMB DEGs and (**e**) 181 PFC DEGs. One-sided *p*-values for Modified Fisher's Exact test are defined. PCA, Principal component analysis; PFC, prefrontal cortex; vMB, ventral midbrain; DEGs, differentially expressed genes; GO, Gene Ontology; DAVID, Database for Annotation, Visualization and Integrated Discovery.

To conclude, we demonstrate that a novel mutation in *CDH2* is associated with familial ADHD, through impaired presynaptic vesicle clustering, attenuated evoked transmitter release, decreased spontaneous release, and reduction in dopaminergic distribution within limbic pathways. We thus delineate the role of *CDH2*-related pathways in the pathophysiology of ADHD.

## Methods

**Subjects and clinical phenotyping**. Nine affected individuals of consanguineous Bedouin kindred were studied. DNA samples were obtained following informed consent and approval of the Soroka Medical Center Internal Review Board (IRB). Clinical phenotyping was determined by an experienced team of pediatric neurologists, psychiatrist, and geneticists for all affected individuals, their parents, and siblings.

**Linkage and sequence analysis**. Genome-wide linkage analysis of all family members and whole-exome sequencing (WES) of an affected individual were performed. WES data were analyzed using QIAGEN's Ingenuity Variant Analysis software (http://www.qiagenbioinformatics.com/ingenuity-variant-analysis; QIAGEN Redwood City, California, USA), excluding variants observed with an allele frequency ≥1% in the 1000 Genomes Project (http://www.internationalgenome.org/) or the Genome Aggregation Database (gnomAD, https://gnomad.broadinstitute.org/), or variants appearing in a homozygous state in our in-house WES database of 400 controls. Of the remaining variants, we selected only those segregating within the family as expected for autosomal recessive heredity.

***CDH2* multiple sequence alignment**. Eight representative CDH2 orthologs were selected for multiple sequence alignment (MSA). All protein sequences were taken from the National Center for Biotechnology Information GenBank (http://www.ncbi.nlm.nih.gov). The RefSeq sequence accession numbers for *H. Sapiens, P. Troglodytes, P. Abelii, M. Mulatta, B. Taurus, C. Lupus Familiaris, R. Norvegicus,* and *M. Musculus* of CDH2 orthologs used for the analysis are NP_001783.2, XP_016788973.1, NP_001125845.1, XP_014977198.1, NP_001159964.3, NP_001125845.1, NP_112623.1, XP_006525616.1 respectively. Protein multiple sequence alignment was performed using Clustal Omega online program (https://www.ebi.ac.uk/Tools/msa/clustalo/).

**Structural modeling**. N-cadherin protein modeling was predicted using the SWISS-MODEL server[66]. Its structure was based on the crystal structure of protocadherin GAmmaB4 extracellular domain with a GMQE score of 0.43 (PDB ID 6E6B)[67]. Modeling of the docking between furin protease (PDB ID 4Z2A) and the peptide harboring the N-cadherin region of interest, was done using the HPEP-DOCK server, predicting protein-peptide interactions[68,69]. PyMOL program (https://sourceforge.net/projects/pymol/) was used to generate all structural figures[70].

**Peptide synthesis**. WT and mutant 22 amino-acids peptides, harboring the RXK/R-R recognition motif, were synthesized by GL Biochem (Shanghai, China) and conjugated with both FITC and biotin at their N and C-terminus, respectively. WT: FITC-SK*H*SGHLQRQKRDW-K-biotin; Mutant: FITC-SK*Y*SGHLQRQKRDW-K-biotin. Peptides were validated by HPLC and MS at 95% purity and dissolved in ultra-pure water (BI, Israel) to 1 mg/ml.

**Furin protease digestion assay**. Recombinant furin protease was purchased (NEB, #P8077S, Ipswich, MA, USA). Furin reaction mixture included 100 mM HEPES (pH 7.5), 0.5% Triton X-100, 1 mM $CaCl_2$ and 1 mM 2-mercaptoethanol, as suggested by the manufacturer. Each mixture (200 µl) was supplemented with a total of 10 µg dissolved peptides and 2U furin enzyme (2000U/1 ml) followed by incubation at 30 °C. At four time intervals (0, 30, 60 and 180 min), 30 µl aliquots of the reaction mixture were removed and immediately deactivated by 80 °C heating for 15 min. Digests were then subjected to LC-MS analysis on a Q Exactive Focus Mass Spectrometer (Thermo Fisher Scientific, USA) operating with an MS resolution of 70,000. All experiments were performed in triplicates.

**Liquid chromatography-mass spectrometry analysis**. WT and mutant peptide extracts were analyzed using a Thermo Scientific Dionex Ultimate 3000 UPLC system coupled to a Q Exactive Focus mass spectrometer (Thermo Scientific, San Jose, USA). The analysis was performed using affinity separation and electrospray approaches. The samples were injected on the Accucore C18 (2.6 µm, $100 \times 2.1$ mm, Thermo Scientific) column, maintained at 30°C, using a mobile phase consisting of 0.1 % formic acid aqueous solution (v/v) and 0.1% formic acid in Acetonitrile (v/v) at the flow rate of 0.3 mL/min. Initial starting conditions were 15% B (equating to 15% ACN) and held for 1 min. A ramp to 60% B was performed over 8 min, and the column was then washed with 100% B for 3 min before returning to starting conditions for 3 min, totaling an entire run time of 15 min. Electrospray analysis was performed in the positive ionization mode using a spray voltage of 2.5–3.5 kV, the tune settings for the MS used an S-lens setting of 50. A full scan range of 300–2000 m/z was used at a resolution of 70,000 (Supplementary Figure 1).

**Generation of *Cdh2* CRISPR/Cas9-mediated mutant mice**. *Cdh2* KI mice, harboring the human pathogenic mutation (replacement of His150 with Tyr) in the mouse ortholog, were generated using the CRISPR/Cas9 system. sgRNA and Cas9 mRNA were designed using SnapGene (https://www.snapgene.com/) and synthesized in-vitro using the mMESSAGE mMACHINE T7 Ultra kit (Life Technologies, cat. AM1345) and MEGAshortscript T7 kit (Life Technologies, cat. AM1354), respectively. Following RNA purification, the sgRNA (10 ng/µl) and Cas9 mRNA (10 ng/µl) were injected into C57BL/6JRcc mice zygotes in the presence of the ssODN (10 ng/µl) using a microinjection system under standard conditions. The zygotes were cultured in a culture medium at 37 °C in 5% $CO_2$ up to two-cell embryos and then transferred into the oviduct of the recipient mice. After birth, 14-21 days old pups were tailed for genomic DNA extraction using TaKaRa *Ex Taq* kit (Takara Bio Inc.) followed by genotyping. Selected KI F0 mice were grown and

bred with C57BL/6JRcc WT mice for two cycles to generate non-chimeric F1 KI heterozygotes. Heterozygote F1 offspring were then bred and F2 offspring of WT and mutant origin (*Cdh2*[H150T] and *Cdh2*[H150Y(2)] founder lines) were used for all further experiments. All primers and oligos sequences appear in Supplementary Table 5.

**Animal maintenance and ethics statement**. The mouse colony was generated and maintained in the animal facility of the Ben-Gurion University of the Negev on a 12:12 h light/dark schedule with food and water provided *ad libitum* at temperature of 20–24 °C with 30–70% humidity. All experiments were carried out following the National Institutes of Health guidelines for the care and use of laboratory animals (NIH Publications No. 8023, revised 1978) and by the guidelines of the Israeli Council on Animal Care. The study protocol was approved by the Committee on Animal Care and Use of the Ben-Gurion University of the Negev.

**Mice behavioral and cognitive evaluation**. Male 10-week-old WT and *Cdh2*[H150Y] mutant mice ($n = 18$, nine mice per group) underwent a 3-week battery of cognitive and behavioral assessment (Fig. 3). Further behavioral experiments (Fig. 4a-g) were conducted on male 12-week-old C57BL/6JRcc WT (Envigo, Israel) and *Cdh2*[H150Y(2)] mice ($n = 30$, 15 mice per group). Behavioral tests were performed in an SPF-certified examination room at HaddasaBrainLabs (http://brainlabs.org.il). The experimental procedure was according to the ARRIVE guidelines and NIH approval number: OPRR-A01-5011. All experiments were approved by the Hebrew University Ethics Committee on Animal Care and Use (Applications MD-14-14015-4; MD-16-14679-4). As an AAALAC-accredited Institute, the Hebrew University Ethics Committee follows the NRC Guide for the Care and Use of Laboratory Animals. We affirm that we have complied with all relevant ethical regulations for animal testing and research. Behavioral tests were recorded and analyzed using the Ethovision 11 system (Noldus Information Technologies, Wageningen, The Netherlands).

**Behavioral experiments**. Motor domains included the open-field exploratory test (OFT) to assess activity and neophobia and the Rotarod test to assess motor-associated functions such as flexibility, endurance, and balance. Cognitive domains included the spontaneous alternation test (Y-maze) to assess executive functions, especially spatial learning and working/reference memory. Anxiety domains include the elevated plus-maze test (EPM) to assess anxiety-like phenotype by monitoring animal levels of risk avoidance and the Acoustic Startle Reflex test (ASR) to evaluate an animal's level of stress by measuring the extent of auditory tone induced flinching. A pre-pulse inhibition test was conducted in the same chamber as the ASR, evaluating sensorimotor gating in an attempt to identify deficits in early-stage information processing. Social interaction domains included the resident-intruder test to assess for aggressive social interaction and the three-chamber sociability test followed by a social novelty test, evaluating animal levels of sociability.

**Behavioral experiments - detailed**

*Open-field*. The apparatus consisted of a square arena measuring 50X50X33cm under 15 lux illumination. The outer walls were wrapped with white paper to limit external stimuli and light gradients. Mice from both genotypes were allowed to explore the arena for 6 min, while their location was tracked and recorded by a video camera positioned overhead. The time spent in the central zone of the arena ($10 \times 10$ cm) was extracted.

*Elevated plus-maze*. The apparatus consisted of four arms ($30 \times 5$ cm each) extending from a central platform ($5 \times 5$ cm). One set of arms, opposing one another, was enclosed by a 15-cm wall ('closed arms'), while the other set was open with a 1-cm ledge on either side ('open arms'). The maze was elevated 75 cm above the ground with illumination set 15 lux. Mice from both genotypes were placed in the central platform and allowed to explore the maze freely for 6 min. Entry into an arm was scored when the center of mass of the animal had entered an arm. The time spent and entries into each arm were extracted for analysis.

*Y-maze*. A spontaneous alternation test was performed to assess executive functions. The Y-shaped maze consisted of 3 plastic arms placed at 120° angles to each other. Mice were placed at the end of one arm and were allowed to explore the maze freely for 6 min without training, reward, or punishment. An alternation was defined as a complete cycle of consecutive entrances into each of the three arms. Percent alternation (PA) was calculated as follows: PA = number of alternations/(total number of entries into each arm – 2).

*Resident-intruder*. A test for aggressive social interaction. The intruder mouse is introduced into the cage of the test resident mouse following habituation. The observation starts when the resident first sniffs the intruder. The observation stops when the first attack (by either mouse) occurs, or when no attack has occurred by 5 min observation[71].

*Acoustic startle reflex (ASR)*. This test evaluates an animal's level of stress/arousal by measuring the extent of audible tone-induced flinching after acclimatization to background noise. Startle trials consist of a single noise burst (120 dB, 40 ms), thereafter the amplitude of animal flinch is recorded. Mean startle amplitude is calculated in a fully computerized, blinded, and unbiased measurement.

*Pre-pulse inhibition (PPI) of the ASR*. PPI is a measure of sensorimotor gating used to identify deficits in early-stage information processing. PPI trials consist of a pre-pulse of intensity 2, 4, 8, or 16 dB above background noise followed 100 ms later by a startling pulse (120 dB, 40 ms). Sessions are designed to include acclimation, pre-pulses, and pulse trials.

*Rotarod*. A test of motor abilities, which requires mice to balance on a rotating cylinder. The test consisted of three 4-minute trials. During each trial, rod rotation gradually increased up to 40 rotations/minute. The duration mice were balanced on the rod in each trial was measured. Trials were divided by at least 20-minute breaks, to avoid mice exhausting.

*Three-chamber sociability test and social novelty test*. Aims to evaluate animal levels of sociability (preference of an unfamiliar mouse over an object) and preference for social novelty (preference of a novel stranger over a familiar one). The device used in this test consists of three chambers (left, right, and central); the middle chamber is connected to the others via doors. In the habituation phase, the animal is allowed to explore the device freely for 10 min. In the second sociability phase, a stranger mouse is placed in one of the lateral chambers (inside a specially devised cup with bars). In the social novelty phase, another stranger is similarly introduced into the other lateral chamber. During phases 2 and 3, time spent in each chamber, the number of approaches to each stranger mouse and their frequency are tracked and recorded[72].

**Methylphenidate hydrochloride administration**. Mice were administered with Methylphenidate hydrochloride (Biotechne 2 A/248099, UK)[73]; Methylphenidate (MPH) was dissolved in 0.9% normal saline and diluted to 1 mg/ml. Male 14-week-old C57BL/6JRcc WT (Envigo, Israel) and *Cdh2^H150Y(1/2)* mice (*n* = 39) were weighed and injected with MPH or vehicle (0.9% saline solution) via intraperitoneal injection at 10 mg/kg body weight, 30 min before the initiation of tests. MPH dosages were chosen based on previous studies in rodents suggesting that these MPH dosages mirror those that are used in clinical practice[15]. MPH experiments were approved by the Hebrew University Ethics Committee on Animal Care and Use (Applications MD-20-16347-3).

**Primary dense hippocampal cultures**. Primary dense hippocampal cultures from P0-P2 pups of either sex were generated. Briefly, postnatal day 0–2 pups from WT and *Cdh2^H150Y* littermates were decapitated and their brains quickly removed; hippocampi were dissected, sliced manually, and kept on ice in Hank's Balanced Salt Solution (Biological Industries, Bet-Haemek, Israel) supplemented with 20 mM HEPES (termed HBSS; Biological Industries, Beit-Haemek, Israel) at pH 7.4. Hippocampus pieces were incubated for 20 min at room temperature (RT) within a digestion solution consisted of 5 ml HBSS, CaCl$_2$ 1.5 mM, EDTA 0.5 mM and 100 units of Papain (Worthington, Lakewood, NJ) activated with Cysteine (Sigma-Aldrich, Rehovot, Israel). Brain fragments were then gently triturated twice. Cells were seeded at a density of 80,000–100,000 cells per well on 12 mm #1 glass coverslips (CS; Bar-Naor Ltd, Ramat-Gan, Israel) coated with poly-D-Lysine (Sigma-Aldrich, Rehovot, Israel). Initially, cells were plated in a plating medium consisting of Neurobasal-A medium supplemented with 2% B27, 2 mM Glutamax I (Thermo-Fisher Scientific, Waltham, MA), 5% defined FBS and 1 μg/ml gentamicin (Biological Industries, Beit-Haemek, Israel). After 24 h, the plating medium was replaced by a serum-free culture medium consisted of Neurobasal-A, 2 mM Glutamax I, and 2% B27. Cultures were maintained at 37 °C in a 5% CO$_2$ humidified incubator for about 12-15 days prior to staining and imaging.

**Immunocytochemistry of hippocampal cultures**. Days in-vitro (DIV) 8 and DIV 14 hippocampal neurons were fixed with 4% paraformaldehyde (EMS, Hatfield, PA) in PBS for 10 min, rinsed with PBS, permeabilized with 0.1% Triton X-100 in PBS for 2 min, blocked with 5% powdered skim-milk (Sigma-Aldrich, Rehovot, Israel) in PBS for 1 h, rinsed, incubated with the primary antibody for 1 h, rinsed, incubated with the secondary antibody for 1 h, rinsed, and mounted in immu-mount (Thermo Fisher Scientific, Waltham, MA). All steps were performed at RT. Primary antibodies used: rabbit polyclonal anti-Synaptobrevin 2 (1:1000, Synaptic Systems), goat polyclonal anti-vesicular glutamate transporter-1 (vGlut1, 1:1000, Synaptic Systems), mouse monoclonal anti-Glutamic acid decarboxylase-6 (GAD-6, 1:1000, developed by D.I. Gottlieb, obtained from the Developmental Studies Hybridoma Bank, DSHB). Secondary antibodies used: Donkey anti-mouse IgG and donkey anti-rabbit IgG, labeled with Northern Lights 637 or 557, respectively (1:1000, R&D Systems), and donkey anti-goat IgG, labeled with AlexaFluor 647 (1:1000, Abcam).

**Semi-quantitative synaptic immunofluorescence**. To allow for a semi-quantitative comparison of immunostaining intensity of synapses, WT, and mutated *Cdh2^H150Y* hippocampal neurons were processed under identical conditions. Synapses were detected semi-automatically using an in-house iterative algorithm based on serially decreasing to thresholds implemented in NIS-elements software (Nikon). Fluorescence values for each synapse were obtained from an area of 3 × 3 pixels located on its center of mass and an image average was generated. Because intensity values can vary from session to session, a normalization value was determined from the WT experiments of each session and further used to normalize all images acquired during that session, both WT and mutants. The normalized values were either averaged or used to compute cumulative distributions. Puncta that were positive for GAD65 were deemed GABAergic since GAD (Gamma-Amino Decarboxylase) is a vesicular enzyme mediating convergence of glutamate to GABA[74]; otherwise, they were considered glutamatergic.

**Fluorescence microscopy of neuronal cultures**. Fluorescence measurements were performed on a Nikon TiE inverted microscope driven by the NIS-elements software package (Nikon). The microscope was equipped with an Andor Neo 5.5 sCMOS 12 bit camera (Oxford Instruments), a 40 × 0.75 NA Super Fluor objective, a 60 × 1.4 NA oil-immersion apochromatic objective (Nikon), a perfect-focus mechanism (Nikon), and EGFP and Cy3 TE-series optical filter sets (Chroma), as well Cy5 filter set (Semrock).

**Synapse width analyses**. Synapse width was measured by drawing a line starting in the axon and through the synapse punctum, and fitting it using a Gaussian function as follows[75]:

$$y = y_0 + A e^{-\frac{(x - x_c)^2}{2w^2}} \qquad (1)$$

Where $x_c$ is the center of the punctum maximum, $y_0$ is the fluorescence of the axon, $w^2$ is the variance of the Gaussian, and $A$ is its amplitude. The full width at half-maximum (FWHM) fluorescence intensity is calculated as follows:

$$FWHM = 2w\sqrt{\ln(4)} \qquad (2)$$

Fitting was performed by the least-squares error method using Origin 2020 (OriginLab). Independent images were acquired from cells grown on various coverslips obtained from a minimum of three different cultures. A mean FWHM was determined for each image, and then a grand average was calculated for WT and mutant cultures.

**FM dye loading and unloading**. Neurons were initially placed in normal saline (in mM: 150 NaCl, 3 KCl, 10 HEPES, 2 CaCl$_2$, 2 MgCl$_2$, 20 Glucose, pH adjusted to 7.35 with NaOH), then loaded with FM1-43 at a final concentration of 10 μM by depolarizing them for 2 min using hyperkalemic saline (90 KCl, 63 NaCl, otherwise as above) in the presence of the dye. The neurons were exposed to the dye in normal saline for an additional 5 min after depolarization, to allow labeling related to endocytosis to be completed. Afterward, neurons were washed with normal saline for 5 min, followed by 5 min washing with 1 mM ADVASEP-7[76] in normal saline. Finally, the cells were washed in normal saline and imaged. The cells were stimulated twice, once for 2 s at 20 Hz to access the size of the RRP and a second time for 120 sec to access the RcP, following a minute of recovery. The degree of unloading was calculated as the change in fluorescence (ΔF) normalized by baseline background-corrected fluorescence F$_0$. Experiments were performed at RT in the presence of APV (50 μM) and DNQX (10 μM) to reduce destaining due to spontaneous network activity[77].

**Measuring vesicle cycling using sypHy**. SypHy is a probe based on the internal fusion of a pH-sensitive GFP called pHluorin[78] (pKa = 7.6), with the vesicular protein Synaptophysin I (SypI), so that pHluorin is located in the lumen of the synaptic vesicles (SVs) and allows the reportage of presynaptic activity[31,79]. Briefly, pHluorin fluorescence is quenched by the acidic pH of the intact vesicle lumen (pH~5.5). Upon stimulation, SVs fuse with the plasma membrane, thus exposing their lumen to the neutral pH of the extracellular environment (pH~7.3), causing an increase in the fluorescence of pHluorin. Following endocytosis, pHluorin is re-quenched due to the reacidification of the lumen by the vesicular proton pump (vATPase). Therefore, an increase in fluorescence reflects the exocytosis phase, and the subsequent decrease in fluorescence indicates the endocytosis phase of the vesicle cycle. The kinetics of exocytosis is determined by examining the time course of an upsurge in fluorescence upon stimulation in the presence of bafilomycin A, an inhibitor of the vATPase[80], which inhibits neurotransmitter loading but not vesicle recycling[81,82]. In our experiments, 12-14 DIV neurons were field-stimulated in the presence of APV (50 μM) and DNQX (10 μM), in the presence or absence of bafilomycin A (Fig. 6a-c). After stimulation, the bath was perfused with saline in which 50 mM NaCl was replaced with NH$_4$Cl. Ammonium ions are in equilibrium with aqueous ammonia, which is membrane-permeable and can diffuse into SVs to neutralize their lumen. Therefore, the combination of sypHy and NH$_4$Cl reveals intact SVs within the terminals, and the size of the total SVs pool is measurable. The recycling vesicle pool (RcP) size is estimated by measuring the plateau of

fluorescence intensity obtained during exhaustive stimulation (Fig. 6c), while the resting pool is that which completes the RcP to the total pool[26]. The baseline fluorescence intensity of sypHy (F0) in each synapse of interest was the average value measured in five successive images acquired before stimulation. The change in fluorescence (ΔF) at time t was calculated as F(t)−F0. Values for each synapse were normalized by the maximal fluorescence intensity (Fmax) measured after the treatment with $NH_4Cl$ (ΔF/Fmax).

**Acute hippocampal slices.** P18-P21 littermate mice for field-potential recordings and P28-P31 mice for whole-cell recordings from either sex were anesthetized with Isoflurane and decapitated. The brains were rapidly removed and placed in an ice-cold oxygenated cutting solution that contains (in mM): 252 Sucrose, 5 KCl, 1 $CaCl_2$, 3 $MgSO_4$, 26 $NaHCO_3$, 1.25 $NaH_2PO_4$, and 10 Glucose; pH 7.3 when bubbled with 95% $O_2/CO_2$. Transverse slices (300 μm) were cut on a vibratome (Leica 3000) and placed into a holding chamber containing oxygenated aCSF at RT for at least an hour prior to the recordings. ACSF solution contains (in mM): 124 NaCl, 3 KCl, 2 $CaCl_2$, 2 $MgSO_4$, 1.25 $NaH_2PO_4$, 26 $NaHCO_3$ and 10 glucose; pH 7.4 when bubbled with 95% $O_2/CO_2$.

**Whole-cell recordings.** Slices were viewed through 40X water-immersion lenses (Olympus) in a BX51WI microscope (Olympus) mounted on an X–Y translation stage (Luigs and Neumann). Somatic whole-cell recordings were made using patch pipettes pulled from thick-walled borosilicate glass capillaries (1.5 mm outer diameter; Science Products). Pipettes had resistances of 5–7 MΩ when filled with a whole-cell voltage-clamp solution that contains (in mM): 135 CsCl, 4 NaCl, 2 $MgCl_2$ and 10 HEPES (cesium salt), pH adjusted to 7.3 with CsOH. Voltage-clamp recordings were made with a MultiClamp 700B amplifier (Molecular Devices). Data were sampled at 10 kHz, amplified (gain 5), filtered at 3 kHz then digitized and analyzed using Clampfit. Membrane access resistance was maintained as low as possible (5-10 MΩ) and was compensated at 80%. Recordings were not corrected for liquid junction potential. Recordings were performed at 30 °C in the presence of Tetrodotoxin (TTX, 1 nM, Alomone Labs) and Bicuculline (GABA A antagonist. 10 μM, Alomone Labs). Inter-event intervals (IEIs) were calculated per recording (per cell) and averaged across slices (up to 3 slices per mouse).

**Two-photon reconstruction of neuronal morphology.** At the end of physiological recordings brain slices from WT and $Cdh2^{H150Y}$ littermates, the slice that contained the cells filled with SBFI fluorescent dye (0.5 μM, Thermo Fisher) was placed in an aCSF perfused chamber. Slices were scanned with an Ultima IV two-photon microscope (Bruker) equipped with a Mai Tai DeepSee pulsed laser (Spectra-Physics) and Olympus water-immersion lens ×60. Using two-photon excitation at 740 nm, we focused on the cell and scanned 30–40 slices of dendrites, soma, and axon at 0.5-μm depth intervals. The resulting z-stacks were imported into ImageJ (US National Institutes of Health) for reconstruction and image processing.

**Field-potential recordings.** Extracellular stimulation was delivered using a stimulus isolation unit (A.M.P.I) using glass monopolar electrodes (0.5–1 MΩ) filled with aCSF. fEPSPs were recorded in current-clamp mode with a MultiClamp 700B amplifier (Molecular Devices) driven by the Clampex (pClamp 10.0) software (Molecular Devices) using aCSF-filled patch pipettes (0.5–1 MΩ). The stimulating electrode was placed in the CA3 area and the recording electrode in the dendritic CA1 area. Experiments were performed at 29 ± 1 °C with flow rates of 2 ml per minute. Five consecutive stimulations were delivered at 5, 10, and 20 Hz. Stimulation intensity was set to acquire an initial fEPSP of approximately 0.3 mV. 5–10 sweeps were conducted for each stimulus frequency and recordings were averaged over trials. Data were sampled at 10 kHz, amplified (gain 5), filtered at 3 kHz then digitized and analyzed using Clampfit. Data were included if the stimulation intensity to fEPSP ratio was within the linear range and if there was an increase in the amplitude of the second response. Data were averaged per slice, and then across the number of specified slices. Up to 3 slices were usable per mouse.

**Presynaptic cytoplasmatic calcium measurements.** Neurons infected with hSyn:SypI-GCaMP6f were imaged at 12-14 DIV during field-stimulation in the presence of APV (50 μM) and DNQX (10 μM) to block recurrent network activity. Field stimulation (20 action potentials at 20 Hz) was applied at an intensity of 10 V/$cm^2$. Experiments were imaged at a sampling rate of 37 Hz, 3 × 3 binning, using an EGFP filter set (Chroma) and a Neo 5.5 Andor sCMOS camera (Oxford Instruments).

**Mouse brain dissections for immunofluorescence and RNA-seq analysis.** After performing a battery of behavioral tests, all animals examined (at 13 weeks) were perfused transcardially with cold phosphate-buffered saline (PBS) followed by 4% formaldehyde, and brains were quickly removed and separated into two hemispheres. We randomly placed one hemisphere in 4% formaldehyde for immunofluorescence analyses and the other was frozen in liquid nitrogen. After 24 h, the hemispheres for immunofluorescence studies were placed in a 30% sucrose solution in DDW and then frozen at optimal cutting temperature. Hemispheres were dissected to 50-micron frozen floating sections, (taken from −2.92 to −3.4 millimeters relative to bregma). Five sections were analyzed for each mouse, with a total of 18 animals in the analysis (nine mice per group). The second hemisphere of each brain was further micro-dissected, separating the prefrontal cortex (PFC) and ventral midbrain (vMB) for real-time quantitative PCR (qPCR) and whole-transcriptome RNA-seq analyses.

**Real-time quantitative PCR (qPCR).** Total RNA was extracted from the micro-dissected tissues using GENzol$^{TM}$ Tri RNA Pure Kit (Geneaid Biotech Ltd.) according to the manufacturer's instructions. Single-stranded cDNA libraries were prepared using Verso cDNA synthesis Kit (Thermo Scientific). qPCR was done using a FastStart Universal SYBR Green reaction master mix (Roche). The analysis was performed using a Rotor-Gene RG-3000 machine (Corbett Research). Tyrosine hydroxylase (TH) and dopamine transporter (DAT) transcript levels in each of the two dissected anatomical areas were normalized to the murine 18 S rRNA housekeeping gene. The annealing temperature used was 60 °C, extension time was set for 20 s and the PCR reaction repeats were 40 cycles. The threshold cycle and double-delta CT were determined manually. All primer sequences used for amplification appear in Supplementary Table 5.

**Tyrosine hydroxylase immunofluorescence visualization.** Brain sections were fixed in methanol. Following three PBS washes, sections were incubated overnight in 1% bovine serum and 0.1% triton in PBS with the primary antibody (mouse anti-Tyrosine hydroxylase 1:50, sc-25269, Santa Cruz Bio.) at 4 °C. Sections were then incubated with the secondary antibody (Alexa Fluor 488, Cy2-donkey anti-mouse, 1:400; Jackson, ImmunoResearch) for 2 h at RT and counter-stained with DAPI (Sigma, Israel).

**Immunofluorescence image analysis.** TH-positive cells' images were captured using an Olympus FV-1000 confocal microscope and camera (Tokyo, Japan). The number of TH-marked cells was manually counted at x10 magnification in a defined area containing the vMB. The number of visible TH-stained cells was divided by the area volume. The area volume was calculated as follows: (area * number of stacks * stack spacing * μm/pixels$^2$)/10$^9$. All histological staining and visualization were performed by a researcher who was blind to the experimental conditions of the specimens. The experimental condition of each specimen was exposed only during analysis.

**Mice brain dissections for quantifying dopamine levels.** Male 14-week-old C57BL/6JRcc WT (Envigo, Israel) and $Cdh2$ mutant mice (from both founder lines, $n = 20$, 10 mice per group) were used for dopamine concentration analysis. Brains were dissected as aforementioned, with two sections analyzed per mouse: PFC and vMB. The micro-dissected tissues were homogenized in 300 μl of dopamine stabilizer buffer (Eagle Biosciences inc. by DLD diagnostika) and kept according to the manufacturer's instructions.

**Measuring dopamine levels.** Dopamine concentration in PFC and vMB tissue samples was calculated using the dopamine high sensitivity enzyme-linked immunosorbent assay (ELISA) kit (Eagle Biosciences inc. by DLD diagnostika). In brief, this competitive assay uses a microtiter plate format with tissue dopamine bound to the solid phase of the plate. Acylated catecholamine from the sample and solid phase bound catecholamine to compete for a fixed number of antiserum binding sites. When the system is in equilibrium, free antigen and free antigen-antiserum complexes are removed by washing. The antibody bound to the solid phase catecholamine is detected by anti-rabbit IgG/peroxidase. The substrate reaction was monitored at 450 nm with a 570 nm reference wavelength (Tecan infinite$^®$ M1000). The amount of antibody bound to the solid phase catecholamine is inversely proportional to the catecholamine concentration of the sample.

**RNA-sequencing whole-transcriptome analysis.** Raw reads were obtained by sequencing on Illumina's NextSeq 500, single end 60. Raw reads were pre-processed by trimming Illumina adapters (GATCGGAAGAGCACACGTCTG AACTCCAGTCAC) and also by trimming edges of reads with quality below 10 using Cutadapt v1.8.3[83]. Reads with lengths below 40 were discarded. Next, reads that contained more than 50% A or 50% T were also discarded. The processed reads were mapped to the human genome, GRCh38, using TopHat v2.0.10[84] and then the number of reads that were mapped to each gene were calculated using HTSeq-count, v0.6.1p1[85]. The gene annotation used for the counting was Ensemble annotation for GRCh38, release 83. Differential expression was assayed using Deseq2[86], comparing samples that were $Cdh2$-mutated using CRISPR/Cas9 versus WT. Genes demonstrating absolute log fold change greater than or equal to 1.3, as well as adjusted $p$-value lower than or equal to 0.05, were considered as differentially expressed. Adjusted $p$-values were taken from Deseq correction for multiple testing, which is based on Benjamin-Hochberg FDR[87]. Genes that were differentially expressed in both cases were considered for further analysis.

**Gene ontology analysis.** To analyze disease-associated enrichment of differentially expressed genes, we used the DAVID functional annotation tool[88,89]. We focused only on genes with adjusted $p$-values lower than or equal to 0.05 and with a

fold change greater than or equal to 1.3. The entire list of genes in the Mus musculus database was used as a reference list and the statistics for both analyses were done without Bonferroni correction (while Bonferroni correction decreases the likelihood of false positives, it increases the likelihood of having type II errors – false negatives, so that truly important terms are sometimes "wrongly" deemed insignificant)[90]. A heatmap of differentially expressed genes was generated using Heatmapper (http://www.heatmapper.ca/). The average linkage method was used for hierarchical clustering and Pearson's coefficient was used for computing the distances between rows and columns.

**Statistics**. Neuronal hippocampal experiments: values are reported throughout as mean ± SEM. Statistical analysis was performed on experiments acquired from at least three independent cultures from WT and $Cdh2^{H150Y}$ mice. Comparisons of two datasets were performed by the Student's $t$-test, after confirming a normal distribution by the Shapiro-Wilk normality test, or with the Mann-Whitney's non-parametric test otherwise. Significance was set at a confidence level of 0.05. Behavioral experiments: univariate differences between the experimental groups were tested using independent student's $t$-test. Multivariate models employed mixed models for repeated measures. TH cell count: as several sections were quantified per mouse, marginal means of control and mutant TH-positive cell density were contrasted using mixed-effects maximum-likelihood regression (Stata 14, USA), with genotype as a fixed effect and individual mouse as a random effect. As directionality was pre-hypothesized based on earlier convergent data from qPCR, a one-tailed 0.05 significance threshold was set. Dopamine concentrations using ELISA: results were provided with 4 Parameter Logistic (4PL) using MyCurveFit Excel add-in (at https://myassays.com/home.aspx). In all figures ns denotes $p > = 0.05$, * denotes $p < 0.05$, **$p < 0.01$, and ***$p < 0.001$. Statistical analysis was performed with OriginPro 2020 (https://www.originlab.com/2020) or SPSS 18 (IBM, Somers, NY).

**Reporting summary**. Further information on research design is available in the Nature Research Reporting Summary linked to this article.

## Data availability

All data that support the findings of this study, including the whole-exosome sequencing data, are available from the corresponding author upon request (O.S.B). The raw and processed mouse RNA-sequencing data generated in this study have been deposited in the Gene Expression Omnibus database under accession code GSE182698. Other data generated in this study are provided in the Supplementary Information and Source Data file. Source data are provided with this paper.

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

## Acknowledgements

We thank the families participating in the study. This research was funded by the Israel Science Foundation (grant no. 2034/18) awarded to Prof. Ohad S. Birk and (grant no. 1310/19) awarded to Dr. Daniel Gitler. The studies were supported in part by the Morris Kahn Family Foundation and the National Knowledge Center for Rare/Orphan Diseases of the Israel Ministry of Science, Technology and Space, Ben-Gurion University of the Negev, Beer-Sheva, Israel. Behavioral experiments and histological studies were performed at HadassahBrainLabs – National Knowledge Center for Research on Brain Disorders, Hebrew University Medical Center, Jerusalem, Israel. We thank Prof. Bernard Lerer, Director of the Biological Psychiatric Laboratory at Hadassah-Hebrew University Medical Center.

## Author contributions

Genetic and molecular studies: DH, RK, MD, OW, OSB, YY, VD, RP, YP. Clinical characterization: GM, DH, OSB. Structural protein analysis: DH, HN. Ex-vivo cleavage assay: DH, BR. Generation of mutant mice: DH. Neurological studies of mice: DH, AS, OS, DG. Behavior studies of mice: DH, TL, AL. OSB initiated and supervised the project. Writing the paper: mostly DH, OSB, DG – approved by all authors.

## Competing interests

The authors declare no competing interests.
