## [Peer Review File · Nature Communications]

CDH2 mutation affecting N-cadherin function causes attention-deficit hyperactivity disorder in humans and miceREVIEWER COMMENTS

Reviewer #1 (Remarks to the Author):

This study identifies an ADHD-related mutation in the CDH2 gene and studies the impacts of this mutation in mice using various approaches, including behavior, synapse, and transcriptome.

I would like to speak highly of the wide range of approaches that the authors have taken and also the quality of the data from the performed experiments. Given the scarcity of reports exploring the biology of ADHD in the field, this manuscript is meaningful and provides sufficient insights into the mechanisms underlying ADHD associated with CDH2.

Major comments

1. The synaptic analyses performed mainly use visualization and cell biological experiments, which are of high quality and draw many useful conclusions. However, confirming these results using electrophysiological experiments would much strengthen the manuscript. For instance, it is unclear what the decrease in the size of synaptic vesicle cluster means in terms of presynaptic release; does it mean a decrease in spontaneous or evoked release of neurotransmitters? Would it change the paired-pulse ratio, or readily releasable pool size? Would the change be specific for excitatory or inhibitory synapses?

2. The behavioral analyses are comprehensive. In particular, the increased hyperactivity is meaningful given the symptoms/phenotypes of ADHD being increased locomotion both in humans and mice. Because ADHD is treated by several medications including methylphenidate and amphetamine, treating the *Chd2*-mutant mice with one of these medications and obtaining some positive rescue results in synaptic and behavioral phenotypes would strengthen the manuscript and help this mouse line properly model ADHD.

3. The decrease in TH levels in the ventral midbrain is interesting. Then it would be important to measure DA levels in their target regions or in the whole brain by experiments such as ELISA or microdialysis.

Minor comments

1. What might be the reason for the strong transcriptomic change in the ventral midbrain but not in the prefrontal cortex?

2. Typo in Fig. 3K.

Reviewer #2 (Remarks to the Author):

Halperin et al. present strong evidence that a missense mutation in the cell adhesion molecule N-cadherin (*cdh2*) leads to ADHD in a single family. They then generate a corresponding mutation in mouse *cdh2*, and show that the mice exhibit behavioral defects that vaguely resemble ADHD, as well as synaptic defects that could underlie the behavioral defects. They finish with an initial RNAseq study that provides candidates for future mechanistic analysis. Altogether, this is a potentially important paper.

However, results from the mutant mouse line need to be strengthened.

1) The most serious problem is with the claimed decrease in dopaminergic signal (Figure 5). The dramatic decrease in immunofluorescent signal in Figure 5C is totally inconsistent with the barely statistically significant difference shown in Figures 5A, B, D and E. Something needs to change. And if the quantification is to be believed, the conclusions in the text should be qualified if retained at all.

2) Because these and also behavioral differences are quite modest, it is important to know more

about how the mutants were prepared and what controls were used. For mutants, how many lines were generated and used, and if more than one, how similar were results? Were mouse outcrossed to wild-type to attenuated possible effects of off-target mutations? The methods section indicates that F2 mice were used. This is likely insufficient. Were the controls wild-type littermates of the mutants? In that the effects are small, it is essential to control for background effects, and this would be the best way to do it.

Specific comments:

Calling ADHD the “most heritable” of behavioral disorders is a strong statement, that requires better documentation.

Another potential mechanism is that cadherins, including *cdh2*, can interact with other cell surface proteins that act as positive regulators of synaptogenesis in specific neuronal subsets (e.g., Yamagata et al., Cadherins Interact with synaptic organizers to promote synaptic differentiation. *Front Mol Neurosci.* 2018). Mentioning this could enhance the scope of the discussion.

Several papers have implicated *cdh13* in ADHD. Those results should be discussed in slightly more detail (p.14).

Reviewer #3 (Remarks to the Author):

In this work, the authors discovered a novel ADHD-associated homozygous missense mutation in *CDH2* gene. CRISPR/Cas9-mediated knock-in mice harboring the equivalent gene mutation recapitulated hyperactivity features and displayed deficient sensorimotor integration. Cultured neurons of mutant mice exhibited impairment of presynaptic vesicle clustering and attenuated synaptic release. Brains of mutant mice showed and reduction in tyrosine hydroxylase distribution within ventral midbrain and PFA and altered gene expression in these regions, implying that some relevant downstream molecular pathways were affected. In general, this work reported an novel disease-causing mutation of *CDH2* gene, and presented interesting experimental findings about molecular and cellular alterations associated with this gene mutation in animal model and neuronal culture. The impact of this work was compromised by the fact that although every pieces of data are very interesting, experimental evidences are loosely related to each other and cannot be integrated together to form a coherent theme. In other word, my feeling is that the breadth of the work sacrificed its depth, leaving some important questions not addressed very well. Some major concerns are listed below.

1. The authors predicted that the mutation may interfere with proteolysis and maturation of the protein. Readers expect to see experimental evidence to confirm this interesting finding. It is important because it explains why this variant is causative to the disease.
2. The authors found that the *Cdh2* mutation weakens synaptic transmission by reducing the size of the SV cluster within the presynaptic terminal without affecting other properties of the synapse, including calcium dynamics and the kinetics of SV usage and recycling using imaging based methodology. Can authors provide more direct evidence using quantitative electronic microscopy? It is better to have more solid evidence presynaptic alteration caused by the mutation. Again, can some of these findings be validated in brain slice, not merely in neuronal culture?
3. The authors reported multiple cellular and molecular alterations including reduced size of SV cluster, reduced TH-neurons and altered gene expression. What is the relationship between these phenotypes? Do they occur in parallel or affect each other in the mutant brain? What is the primary cause of the observed behavioral impairments of the mutant animal? Mechanistically, how *CDH2* mutation may lead to these molecular and cellular alterations? All these important questions have not been addressed in the manuscript.

4. The reduced TH-positive cells in mutant mice is very interesting. How does it occur, increased cell death or reduced genesis of this specific cell population during development, abnormal migration and final location of these neurons, or merely reduced TH gene expression? These are basic questions that need to be addressed to get a clear picture of this cellular phenotype.

CDH2 Article – Reviewers' comments:

We thank the editor and the reviewers for their helpful comments. Our responses to the comments are itemized below.

Reviewer 1:

This study identifies an ADHD-related mutation in the *CDH2* gene and studies the impacts of this mutation in mice using various approaches, including behavior, synapse, and transcriptome. I would like to speak highly of the wide range of approaches that the authors have taken and also the quality of the data from the performed experiments. Given the scarcity of reports exploring the biology of ADHD in the field, this manuscript is meaningful and provides sufficient insights into the mechanisms underlying ADHD associated with *CDH2*.

Major comments:

1. The synaptic analyses performed mainly use visualization and cell biological experiments, which are of high quality and draw many useful conclusions. However, confirming these results using electrophysiological experiments would much strengthen the manuscript. For instance, it is unclear what the decrease in the size of synaptic vesicle cluster means in terms of presynaptic release; does it mean a decrease in spontaneous or evoked release of neurotransmitters? Would it change the paired-pulse ratio, or readily releasable pool size? Would the change be specific for excitatory or inhibitory synapses?

RESPONSE: As the reviewer rightfully pointed out, the total size of the vesicle cluster does not provide direct information concerning synaptic properties. Therefore, as requested, additional imaging and electrophysiological experiments were conducted to address this issue. Specifically, we added extracellular recordings performed from the CA1 area in the hippocampus in brain slices, which show that frequency facilitation and the paired-pulse ratio are smaller in *Chd2* mutant mice, which could arise due to a decrease in the per-vesicle release probability or a smaller releasable pool (leading to enhanced depletion). Furthermore, we performed intracellular recordings of spontaneous release showing that the quantal content is unchanged, but that the frequency of the events is lower in *Chd2* mutant mice. Finally, we performed FM1-43 experiments showing that the readily releasable pool (RRP) in cultured neurons from *Chd2* mice is smaller. We agree that

studying synaptic properties separately in glutamatergic and GABAergic neurons is of interest; however, we feel that such studies are beyond the scope of the current manuscript and should be dealt with in future publications.

2. The behavioral analyses are comprehensive. In particular, the increased hyperactivity is meaningful given the symptoms/phenotypes of ADHD being increased locomotion both in humans and mice. Because ADHD is treated by several medications including methylphenidate and amphetamine, treating the *Cdh2*-mutant mice with one of these medications and obtaining some positive rescue results in synaptic and behavioral phenotypes would strengthen the manuscript and help this mouse line properly model ADHD.

RESPONSE: To further establish and validate the *Cdh2* mouse model, we performed the open-field exploratory test on a different founder line of homozygous mutant KI mice. Congruent with the *Cdh2*^{H150Y(1)} results, 12-week-old male *Cdh2*^{H150Y(2)} mice (n=30, 15 mice per group) further exhibited significantly greater traveling distance, increased velocity, prolonged mobility time and a significant increase in the number of center zone alternation (Fig. 4 A-G) compared to C57BL/6JRcc WT mice (Envigo, Israel, Fig. 4 A-G). Then, as requested, we performed a Methylphenidate hydrochloride (MPH) intervention assay (see methods). In brief, MPH was dissolved in 0.9% normal saline and diluted to 1mg/ml. Male 14-week-old C57BL/6JRcc WT (Envigo, Israel) and *Cdh2*^{H150Y(1/2)} mice (from both founder lines) were acutely administered with MPH or vehicle (0.9% saline solution) via intraperitoneal injection at 10mg/kg body weight, 30 min before initiation of tests. We specifically investigated MPH's effect on explorative behavior and sensorimotor gating, through OFT and ASR, respectively. While both WT and mutant mice exhibited increased locomotor activity following MPH administration, the effect was significantly greater in the *Cdh2* mutant mice (Fig. 4I). In addition, the fold-change of the cumulative duration within center zone of the mutant mice was significantly smaller compared with the WT mice. These results further highlight the in-vivo pathogenicity of the *Cdh2* mutation, with the mutant mice being significantly more susceptible to stimulatory intervention following MPH acute administration.

3. The decrease in TH levels in the ventral midbrain is interesting. Then it would be important to measure DA levels in their target regions or in the whole brain by experiments such as ELISA or microdialysis.

RESPONSE: As requested, we measured DA levels in two mesocortical regions (prefrontal cortex (PFC) and ventral midbrain (vMB)) of WT and *Cdh2*-mutant mice. Brains of male 14-week-old C57BL/6JRcc WT (Envigo, Israel) and *Cdh2*^{H150Y} mice (n=20, 10 mice per genotype) were micro-dissected. We quantified dopamine levels using enzyme-linked immunosorbent assay (ELISA, Eagle Biosciences inc. by DLD diagnostika) in both regions. In line with the altered TH-expression levels in the PFC, we found dopamine concentrations to be significantly decreased in *Cdh2*^{H150Y} mice (Fig. 7F). This further suggests that dopamine tone (via the mesocortical pathway) is affected by the N-cadherin mutation.

Minor comments:

4. What might be the reason for the strong transcriptomic change in the ventral midbrain but not in the prefrontal cortex?

RESPONSE: As demonstrated in figure 7A, the strong transcriptomic change was in fact in the prefrontal cortex (p=0.03), more prominent than the changes observed within the ventral midbrain. This was further supported by measuring dopamine concentrations directly within these two mesocortical domains. Notably, variability in the expression of CDH2 may be directly related to different stages of brain development, specifically synaptogenesis.

5. Typo in Fig. 3K.

RESPONSE: Thank you. Corrected (Fig. 3K).

Reviewer 2:

Halperin et al. present strong evidence that a missense mutation in the cell adhesion molecule N-cadherin (*cdh2*) leads to ADHD in a single family. They then generate a corresponding mutation in mouse *cdh2*, and show that the mice exhibit behavioral defects that vaguely resemble ADHD, as well as synaptic defects that could underlie the behavioral defects. They finish with an initial RNAseq study that provides candidates for future mechanistic analysis. Altogether, this is a potentially important paper. However, results from the mutant mouse line need to be strengthened.

1. The most serious problem is with the claimed decrease in dopaminergic signal (Figure 5). The dramatic decrease in immunofluorescent signal in Figure 5C is totally inconsistent with the barely statistically significant difference shown in Figures 5A, B, D and E. Something needs to change.

And if the quantification is to be believed, the conclusions in the text should be qualified if retained at all.

RESPONSE: Indeed, the conclusion regarding dopaminergic tone based solely on TH-expression levels required further validation and investigation. Following this remark, we decided to directly measure dopamine concentration in two mesocortical domains (prefrontal cortex (PFC) and ventral midbrain vMB): we quantified dopamine levels using enzyme-linked immunosorbent assay (ELISA, Eagle Biosciences inc. by DLD diagnostika) of male 14-week-old WT (Envigo, Israel) and *Cdh2*^{H150Y} mice (n=20, ten mice per genotype, elaborated in the Methods sections). In line with the altered TH-expression levels, we found dopamine concentrations to be significantly decreased in the PFC of *Cdh2*^{H150Y} mice (p=0.013), congruent with the qPCR expression results (Fig. 7F, G). TH-expression level differences are indeed modest in vMB, and as directionality was pre-hypothesized based on earlier convergent data from qPCR, a one-tailed 0.05 significance threshold for the IF experiment was set (Fig. 7C). Nonetheless, an informative IF illustration was chosen (Fig. 7C) to convey this significance. To note, these regions were visualized (rather than PFC) due to technical issues, as these floating sections are much easier to work with, rather than the small PFC fragments. This was clarified and further elaborated in the Results section.

2. Because these and also behavioral differences are quite modest, it is important to know more about how the mutants were prepared and what controls were used. For mutants, how many lines were generated and used, and if more than one, how similar were the results? Were mice outcrossed to wild-type to attenuated possible effects of off-target mutations? The methods section indicates that F2 mice were used. This is likely insufficient. Were the controls wild-type littermates of the mutants? In that the effects are small, it is essential to control for background effects, and this would be the best way to do it.

RESPONSE: Thank you. To further establish and validate the *Cdh2* mouse model, we now performed the open-field exploratory test on a different founder line of homozygous mutant KI mice. Congruent with the *Cdh2*^{H150Y(1)} results, 12-week-old male *Cdh2*^{H150Y(2)} mice (n=30, 15 mice per group) further exhibited significantly greater traveling distance, increased velocity, prolonged mobility time and a significant increase in the number of center zone alternation (Fig. 4 A-G) compared to C57BL/6JRcc WT mice (Envigo, Israel, Fig. 4 A-G).

As requested, we also further elaborated on mice preparation in the Methods section. In brief, selected KI C57BL/6JRcc F0 mice were grown and bred with C57BL/6JRcc WT mice for additional two cycles to generate non-chimeric F1 KI heterozygotes, to attenuate off-target effects. To mention, our sgRNA was chosen based on maximal on-target score, with an ideal off-target score. Heterozygote F1 offspring were then bred and F2 offspring of mutant (*Cdh2*^{H150Y(1)} and *Cdh2*^{H150Y(2)} founder lines) and WT origin were used for further experiments, as now detailed specifically for each experiment. In specific experiments (Second founder line behavioral experiments and MPH intervention), we did not use WT littermates, but rather age and strain-matched controls were purchased instead (Envigo, Israel). In the synaptic experiments, we used both founder lines of the *Cdh2* mutants. In specific experiments (as detailed) littermates were used.

Specific comments:

3. Calling ADHD, the “most heritable” of behavioral disorders is a strong statement, that requires better documentation.

RESPONSE: Thank you. we changed the text in the abstract and discussion accordingly.

4. Another potential mechanism is that cadherins, including *cdh2*, can interact with other cell surface proteins that act as positive regulators of synaptogenesis in specific neuronal subsets (e.g., Yamagata et al., Cadherins Interact with synaptic organizers to promote synaptic differentiation. Front Mol Neurosci. 2018). Mentioning this could enhance the scope of the discussion.

RESPONSE: We now cited this article and further elaborated on this publication in the discussion.

5. Several papers have implicated *cdh13* in ADHD. Those results should be discussed in slightly more detail (p.14).

RESPONSE: Done as requested.

Reviewer 3:

In this work, the authors discovered a novel ADHD-associated homozygous missense mutation in *CDH2* gene. CRISPR/Cas9-mediated knock-in mice harboring the equivalent gene mutation recapitulated hyperactivity features and displayed deficient sensorimotor integration. Cultured neurons of mutant mice exhibited impairment of presynaptic vesicle clustering and attenuated

synaptic release. Brains of mutant mice showed and reduction in tyrosine hydroxylase distribution within ventral midbrain and PFA and altered gene expression in these regions, implying that some relevant downstream molecular pathways were affected. In general, this work reported a novel disease-causing mutation of CDH2 gene, and presented interesting experimental findings about molecular and cellular alterations associated with this gene mutation in animal model and neuronal culture. The impact of this work was compromised by the fact that although every pieces of data are very interesting, experimental evidences are loosely related to each other and cannot be integrated together to form a coherent theme. In other word, my feeling is that the breadth of the work sacrificed its depth, leaving some important questions not addressed very well. Some major concerns are listed below.

1. The authors predicted that the mutation may interfere with proteolysis and maturation of the protein. Readers expect to see experimental evidence to confirm this interesting finding. It is important because it explains why this variant is causative to the disease.

RESPONSE: To address the question of whether the p.H150Y mutation interferes with proteolysis and maturation of the protein, we added a biochemical peptide cleavage assay using furin protease, the prototypical proprotein convertase; WT and mutant 22 amino-acids peptides, harboring the RXK/R-R recognition motif, were synthesized by GL Biochem (Shanghai, China). Peptides were conjugated with both FITC and biotin at their N and C-terminus, respectively. Peptide sequences are WT: FITC-SKHSGHLQRQKRDW-K-Biotin and Mutant: FITC-SKYSGHLQRQKRDW-K-Biotin (Peptides were validated by HPLC and MS at 95% purity). Following digestion with furin, peptides were cleaved into two fragments based on the recognition preference of the protease. We demonstrated through liquid chromatography-mass spectrometry (LC-MS) analysis that the proteolytic cleavage of the mutated peptide is substantially decreased compared with the processing of the WT peptide (Fig. 2E, F). Thus, our data support the postulation that replacing histidine with tyrosine debilitates the anchoring of the peptide within the catalytic pocket of furin protease and putatively impairs protein maturation.

2. The authors found that the Cdh2 mutation weakens synaptic transmission by reducing the size of the SV cluster within the presynaptic terminal without affecting other properties of the synapse, including calcium dynamics and the kinetics of SV usage and recycling using imaging-based methodology. Can authors provide more direct evidence using quantitative electronic microscopy?

It is better to have more solid evidence presynaptic alteration caused by the mutation. Again, can some of these findings be validated in brain slices, not merely in neuronal culture?

RESPONSE: While electron microscopy would provide unequivocal information concerning the number of vesicles, its contribution to their functionality would be limited. Furthermore, we felt this was beyond the capabilities of the participating labs. Instead, as itemized in the answers to reviewer 1, we performed additional electrophysiological experiments as well as FM1-43 ones, which provided functional information concerning the synaptic properties of glutamatergic neurons in the mutant mice, including slice recordings, as requested by this reviewer.

3. The authors reported multiple cellular and molecular alterations including reduced size of SV cluster, reduced TH-neurons and altered gene expression. What is the relationship between these phenotypes? Do they occur in parallel or affect each other in the mutant brain? What is the primary cause of the observed behavioral impairments of the mutant animal? Mechanistically, how CDH2 mutation may lead to these molecular and cellular alterations? All these important questions have not been addressed in the manuscript.

RESPONSE: The notion that N-cadherin is important for presynaptic differentiation is well-established, and we assume it to be a cornerstone for all other observed phenotypes. We now demonstrate the *Cdh2*^{H150Y} mutation to cause defects in synaptic properties, mainly presynaptic SV clustering, synaptic attenuation, frequency facilitation and paired-pulse ratio, which are all smaller in mutant mice. Altogether, the electrophysiological phenotypes could arise due to the decrease in per-vesicle release probability or smaller releasable pool as the initial cause. It has been further demonstrated that N-cadherin has an essential role in regulating the proliferation of dopaminergic progenitors within the limbic system. We assume that debilitated synaptogenesis, as demonstrated through several experiments, could impair TH-expression within mesocortical regions, followed by a decrease in dopaminergic tone/distribution. These differences manifest as behavioral changes in mutated CRISPR/cas9-mutated mice.

4. The reduced TH-positive cells in mutant mice is very interesting. How does it occur, increased cell death or reduced genesis of this specific cell population during development, abnormal migration and final location of these neurons, or merely reduced TH gene expression? These are basic questions that need to be addressed to get a clear picture of this cellular phenotype.

RESPONSE: Examining TH-expression levels in mesocortical regions was based on the notion that N-cadherin has an essential role in regulating the proliferation of dopaminergic progenitors within the limbic system. We assume that TH-expression declined due to impaired synaptogenesis. The reviewer is correct in pointing out, as we now mention in the Discussion, that the reason for this reduction can occur through several mechanisms.

REVIEWER COMMENTS

Reviewer #1 (Remarks to the Author):

The authors have done an extensive job addressing all of my comments successfully. My last comment is on the methylphenidate treatment results. This drug usually increases locomotor activity in WT mice and mouse models of ADHD, which is also true for humans. The results in the present study indicate that locomotor activity is increased in both WT and mutant mice, although it seems to be greater in the mutant mice. These results are different from what we would normally expect, although this result does not necessarily exclude the possibility that these mice do not model ADHD considering that some significant portions of ADHD patients do not respond to methylphenidate or amphetamine. However, these results should be clarified in Results and/or Discussion, together with potential reasons for the unexpected results.

Reviewer #2 (Remarks to the Author):

Halperin et al., revised

My original review was generally favorable but I suggested that the authors reconsider their claim about decreased dopaminergic signal, and also provide be sure that off-target mutations had been eliminated from the mutant line (or at least reduced) by outcrossing. Their responses to both criticisms are exemplary. They also responded well to three minor comments. As far as I can tell, they also added substantial new data in response to suggestions from the other two reviewers. This is an interesting and important paper. I congratulate the authors on their accomplishments.

Minor comments

1) The second line is a major strength, but the outcrossing and littermate controls deserve an additional sentence in "Results."

2) Revision of the abstract to include new data made it hard to read. Here is my humble attempt at simplification: "Attention-deficit hyperactivity disorder (ADHD) is a common childhood-onset psychiatric disorder characterized by inattention, impulsivity and hyperactivity. ADHD exhibits substantial heritability, with rare monogenic variants contributing to its pathogenesis. Here we demonstrate familial ADHD caused by a missense mutation in CDH2, which encodes the adhesion protein N-cadherin, known to play a significant role in synaptogenesis; the mutation affects maturation of the protein. In line with the human phenotype, CRISPR/Cas9-mutated knock-in mice harboring the human mutation in the mouse ortholog recapitulated core behavioral features of hyperactivity. Symptoms were attenuated by methylphenidate, the most commonly prescribed therapeutic for ADHD. The mice exhibited impaired presynaptic vesicle clustering, attenuated evoked transmitter release and decreased spontaneous release. Specific downstream molecular pathways were affected in both ventral midbrain and prefrontal cortex, with reduced tyrosine hydroxylase expression and dopamine levels. We thus delineate roles for CDH2-related pathways in the pathophysiology of ADHD.

CDH2 Article – Reviewers’ comments:

We thank the reviewers for their additional helpful comments. Our responses to the comments are itemized below.

Reviewer 1:

The authors have done an extensive job addressing all of my comments successfully. My last comment is on the methylphenidate treatment results. This drug usually increases locomotor activity in WT mice and mouse models of ADHD, which is also true for humans. The results in the present study indicate that locomotor activity is increased in both WT and mutant mice, although it seems to be greater in the mutant mice. These results are different from what we would normally expect, although this result does not necessarily exclude the possibility that these mice do not model ADHD considering that some significant portions of ADHD patients do not respond to methylphenidate or amphetamine. However, these results should be clarified in Results and/or Discussion, together with potential reasons for the unexpected results.

RESPONSE: Thank you. We further elaborated on these findings in the Discussion.

Reviewer 2:

My original review was generally favorable but I suggested that the authors reconsider their claim about decreased dopaminergic signal, and also provide be sure that off-target mutations had been eliminated from the mutant line (or at least reduced) by outcrossing. Their responses to both criticisms are exemplary. They also responded well to three minor comments. As far as I can tell, they also added substantial new data in response to suggestions from the other two reviewers. This is an interesting and important paper. I congratulate the authors on their accomplishments.

Minor comments:

1. The second line is a major strength, but the outcrossing and littermate controls deserve an additional sentence in “Results.”

RESPONSE: Added as requested in the Results section, under: CRISPR/Cas9-mutated knock-in mice generation.

2. Revision of the abstract to include new data made it hard to read. Here is my humble attempt at simplification: “Attention-deficit hyperactivity disorder (ADHD) is a common childhood-onset psychiatric disorder characterized by inattention, impulsivity and hyperactivity. ADHD exhibits substantial heritability, with rare monogenic variants contributing to its pathogenesis. Here we demonstrate familial ADHD caused by a missense mutation in CDH2, which encodes the adhesion protein N-cadherin, known to play a significant role in synaptogenesis; the mutation affects maturation of the protein. In line with the human phenotype, CRISPR/Cas9-mutated knock-in mice harboring the human mutation in the mouse ortholog recapitulated core behavioral features of hyperactivity. Symptoms were attenuated by methylphenidate, the most commonly prescribed therapeutic for ADHD. The mice exhibited impaired presynaptic vesicle clustering, attenuated evoked transmitter release and decreased spontaneous release. Specific downstream molecular pathways were affected in both ventral midbrain and prefrontal cortex, with reduced tyrosine hydroxylase expression and dopamine levels. We thus delineate roles for CDH2-related pathways in the pathophysiology of ADHD.

RESPONSE: Revised as proposed. Thank you.